# SCORE-BASED PULLBACK RIEMANNIAN GEOMETRY

## ABSTRACT

Data-driven Riemannian geometry has emerged as a powerful tool for interpretable representation learning, offering improved efficiency in downstream tasks. Moving forward, it is crucial to balance cheap manifold mappings with efficient training algorithms. In this work, we integrate concepts from pullback Riemannian geometry and generative models to propose a framework for data-driven Riemannian geometry that is scalable in both geometry and learning: score-based pullback Riemannian geometry. Focusing on unimodal distributions as a first step, we propose a score-based Riemannian structure with closed-form geodesics that pass through the data probability density. With this structure, we construct a Riemannian autoencoder (RAE) with error bounds for discovering the correct data manifold dimension. This framework can naturally be used with anisotropic normalizing flows by adopting isometry regularization during training. Through numerical experiments on various datasets, we demonstrate that our framework not only produces high-quality geodesics through the data support, but also reliably estimates the intrinsic dimension of the data manifold and provides a global chart of the manifold, even in high-dimensional ambient spaces.

## 1 INTRODUCTION

Data often reside near low-dimensional non-linear manifolds as illustrated in Figure 1. This manifold assumption (Fefferman et al., 2016) has been popular since the early work on non-linear dimension reduction (Belkin & Niyogi, 2001; Coifman & Lafon, 2006; Roweis & Saul, 2000; Sammon, 1969; Tenenbaum et al., 2000). Learning this non-linear structure, or representation learning, from data has proven to be highly successful (DeMers & Cottrell, 1992; Kingma & Welling, 2013) and continues to be a recurring theme in modern machine learning approaches and downstream applications (Chow et al., 2022; Gomari et al., 2022; Ternes et al., 2022; Vahdat & Kautz, 2020; Zhong et al., 2021).

Recent advances in data-driven Riemannian geometry have demonstrated its suitability for learning representations. In this context, these representations are elements residing in a learned geodesic subspace of the data space, governed by a non-trivial Riemannian structure[1] across the entire ambient space (Arvanitidis et al., 2016; Diepeveen, 2024; Hauberg et al., 2012; Peltonen et al., 2004; Scarvelis & Solomon, 2023; Sorrenson et al., 2024; Sun et al., 2024). Among these contributions, it is worth highlighting that Sorrenson et al. (2024) are the first and only ones so far to use information from the full data distribution obtained though generative models (Dinh et al., 2017; Song et al., 2020), even though this seems a natural approach given recent studies such as Sakamoto et al. (2024); Stanczuk et al. (2022). A possible explanation for the limited use of generative models in constructing Riemannian geometry could lie in challenges regarding *scalability of the manifold mappings*. Indeed, even though the generative models can be trained efficiently, Sorrenson et al. (2024) also mention themselves that it can be numerically challenging to work with their induced Riemannian geometry.

If the manifold mapping scalability challenges were to be overcome, the combined power of Riemannian geometry and state of the art generative modelling could have profound implications on how to handle data in general. Indeed, beyond typical data analysis tasks such as computing distances, means, and interpolations/extrapolations of data points as illustrated in Figures 2a to 2d, a data-driven Riemannian structure also offers greater potential for representation learning and down-

---

[1]rather than the standard $\ell^2$-inner product

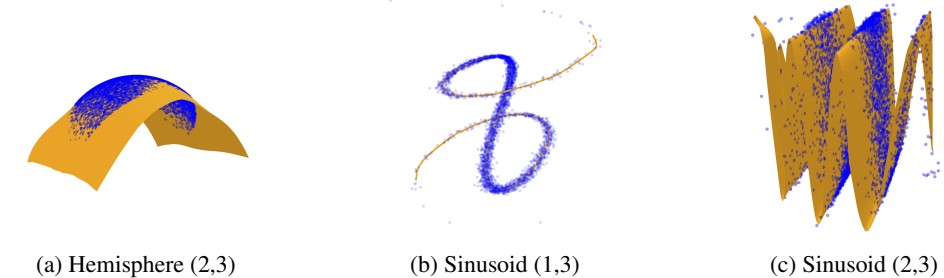

| (a) Hemisphere (2,3) | (b) Sinusoid (1,3) | (c) Sinusoid (2,3) |

Figure 1: Approximate data manifolds learned by the Riemannian autoencoder generated by score-based pullback Riemannian geometry for three datasets. The orange surfaces represent the manifolds learned by the model, while the blue points correspond to the training data. Each manifold provides a convincing low-dimensional representation of the data, isometric to its respective latent space.

stream applications. For instance, many advanced data processing methods, from Principal Component Analysis (PCA) to score and flow-matching, have Riemannian counterparts (Diepeveen et al. (2023); Fletcher et al. (2004) and Chen & Lipman (2023); Huang et al. (2022)) that have proven beneficial by improving upon full black box methods in terms of interpretability (Diepeveen, 2024) or Euclidean counterparts in terms of efficiency (Kapusniak et al., 2024; de Kruiff et al., 2024). Here it is worth highlighting that scalability of manifold mappings was completely circumvented by Diepeveen (2024) and de Kruiff et al. (2024) by using pullback geometry. However, here learning a suitable (and stable) pullback geometry suffers from challenges regarding *scalability of the training algorithm*, contrary to the approach by Sorrenson et al. (2024).

Motivated by the above, this work aims to address the following question: How to strike a good balance between scalability of training a data-driven Riemannian structure and of evaluating its corresponding manifold mappings?

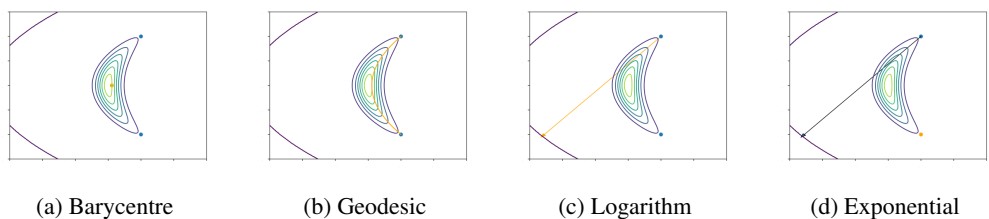

| (a) Barycentre | (b) Geodesic | (c) Logarithm | (d) Exponential |

Figure 2: Proposed Riemannian geometry from a toy probability density, visualized by its level sets.

### 1.1 CONTRIBUTIONS

In this paper, we take first steps towards striking such a balance and propose a score-based pullback Riemannian metric assuming a relatively simple but generally applicable family of probability densities, which we show to result in both scalable manifold mappings and scalable learning algorithms. We emphasize that we do not directly aim to find the perfect balance between the two types of scalability. Instead we start from a setting which has many nice properties, but will allow for generalization to multimodal densities, which we reserve for future work.

Specifically, we consider a family of unimodal probability densities whose negative log-likelihoods are compositions of strongly convex functions and diffeomorphisms. As this work is an attempt to bridge between the geometric data analysis community and the generative modeling community, we break down the contributions in two ways. Theoretically,

- We propose a score-based pullback Riemannian metric such that manifold mappings respect the data distribution as illustrated in Figures 2a to 2d.

- We demonstrate that this density-based Riemannian structure naturally leads to a Riemannian autoencoder[2] and provide error bounds on the expected reconstruction error, which allows for approximation of the data manifold as illustrated in Figure 1.
- We introduce a learning scheme based on adaptations of normalizing flows to find the density to be integrated into the Riemannian framework, which is tested on several synthetic data sets.

Practically, this work showcases how two simple adaptations to the normalizing flows framework enable data-driven Riemannian geometry. This significantly expands the potential for downstream applications compared to the unadapted framework.

## 1.2 OUTLINE

After introducing notation in Section 2, Section 3 considers a family of probability distributions, from which we obtain suitable geometry, and Section 4 showcases how one can subsequently construct Riemannian Autoencoders with theoretical guarantees. From these observations Section 5 discusses the natural limitations of standard normalizing flows and how to change the parametrisation and training for downstream application in a Riemannian geometric setting. Section 6 showcases several use cases of data-driven Riemannian structure on several data sets. Finally, we summarize our findings in Section 7.

## 2 NOTATION

Here we present some basic notations from differential and Riemannian geometry, see Boothby (2003); Carmo (1992); Lee (2013); Sakai (1996) for details.

Let $\mathcal{M}$ be a smooth manifold. We write $C^\infty(\mathcal{M})$ for the space of smooth functions over $\mathcal{M}$. The *tangent space* at $\mathbf{p} \in \mathcal{M}$, which is defined as the space of all *derivations* at $\mathbf{p}$, is denoted by $\mathcal{T}_{\mathbf{p}}\mathcal{M}$ and for *tangent vectors* we write $\Xi_{\mathbf{p}} \in \mathcal{T}_{\mathbf{p}}\mathcal{M}$. For the *tangent bundle* we write $\mathcal{T}\mathcal{M}$ and smooth vector fields, which are defined as *smooth sections* of the tangent bundle, are written as $\mathscr{X}(\mathcal{M}) \subset \mathcal{T}\mathcal{M}$.

A smooth manifold $\mathcal{M}$ becomes a *Riemannian manifold* if it is equipped with a smoothly varying *metric tensor field* $(\cdot, \cdot): \mathscr{X}(\mathcal{M}) \times \mathscr{X}(\mathcal{M}) \to C^\infty(\mathcal{M})$. This tensor field induces a *(Riemannian) metric* $d_{\mathcal{M}}: \mathcal{M} \times \mathcal{M} \to \mathbb{R}$. The metric tensor can also be used to construct a unique affine connection, the *Levi-Civita connection*, that is denoted by $\nabla_{(\cdot)}(\cdot): \mathscr{X}(\mathcal{M}) \times \mathscr{X}(\mathcal{M}) \to \mathscr{X}(\mathcal{M})$. This connection is in turn the cornerstone of a myriad of manifold mappings. One is the notion of a *geodesic*, which for two points $\mathbf{p}, \mathbf{q} \in \mathcal{M}$ is defined as a curve $\gamma_{\mathbf{p}, \mathbf{q}}: [0, 1] \to \mathcal{M}$ with minimal length that connects $\mathbf{p}$ with $\mathbf{q}$. Another closely related notion to geodesics is the curve $t \mapsto \gamma_{\mathbf{p}, \Xi_{\mathbf{p}}}(t)$ for a geodesic starting from $\mathbf{p} \in \mathcal{M}$ with velocity $\dot{\gamma}_{\mathbf{p}, \Xi_{\mathbf{p}}}(0) = \Xi_{\mathbf{p}} \in \mathcal{T}_{\mathbf{p}}\mathcal{M}$. This can be used to define the *exponential map* $\exp_{\mathbf{p}}: \mathcal{D}_{\mathbf{p}} \to \mathcal{M}$ as

$$\exp_{\mathbf{p}}(\Xi_{\mathbf{p}}) := \gamma_{\mathbf{p}, \Xi_{\mathbf{p}}}(1) \quad \text{where } \mathcal{D}_{\mathbf{p}} \subset \mathcal{T}_{\mathbf{p}}\mathcal{M} \text{ is the set on which } \gamma_{\mathbf{p}, \Xi_{\mathbf{p}}}(1) \text{ is defined.} \quad (1)$$

Furthermore, the *logarithmic map* $\log_{\mathbf{p}}: \exp(\mathcal{D}'_{\mathbf{p}}) \to \mathcal{D}'_{\mathbf{p}}$ is defined as the inverse of $\exp_{\mathbf{p}}$, so it is well-defined on $\mathcal{D}'_{\mathbf{p}} \subset \mathcal{D}_{\mathbf{p}}$ where $\exp_{\mathbf{p}}$ is a diffeomorphism.

Finally, if $(\mathcal{M}, (\cdot, \cdot))$ is a $d$-dimensional Riemannian manifold, $\mathcal{N}$ is a $d$-dimensional smooth manifold and $\phi: \mathcal{N} \to \mathcal{M}$ is a diffeomorphism, the *pullback metric*

$$(\Xi, \Phi)^\phi := (D_{(\cdot)}\phi[\Xi_{(\cdot)}], D_{(\cdot)}\phi[\Phi_{(\cdot)}])_{\phi(\cdot)}, \quad (2)$$

where $D_{\mathbf{p}}\phi: \mathcal{T}_{\mathbf{p}}\mathcal{N} \to \mathcal{T}_{\phi(\mathbf{p})}\mathcal{M}$ denotes the differential of $\phi$, defines a Riemannian structure on $\mathcal{N}$, which we denote by $(\mathcal{N}, (\cdot, \cdot)^\phi)$. Pullback metrics literally pull back all geometric information from the Riemannian structure on $\mathcal{M}$. In particular, closed-form manifold mappings on $(\mathcal{M}, (\cdot, \cdot))$ yield under mild assumptions closed-form manifold mappings on $(\mathcal{N}, (\cdot, \cdot)^\phi)$. Throughout the rest of the paper pullback mappings will be denoted similarly to (2) with the diffeomorphism $\phi$ as a superscript, i.e., we write $d_{\mathcal{N}}^\phi(\mathbf{p}, \mathbf{q})$, $\gamma_{\mathbf{p}, \mathbf{q}}^\phi$, $\exp_{\mathbf{p}}^\phi(\Xi_{\mathbf{p}})$ and $\log_{\mathbf{p}}^\phi \mathbf{q}$ for $\mathbf{p}, \mathbf{q} \in \mathcal{N}$ and $\Xi_{\mathbf{p}} \in \mathcal{T}_{\mathbf{p}}\mathcal{N}$.

---

[2]in the sense of Diepeveen (2024)

## 3 RIEMANNIAN GEOMETRY FROM UNIMODAL PROBABILITY DENSITIES

We remind the reader that the ultimate goal of data-driven Riemannian geometry on $\mathbb{R}^d$ is to construct a Riemannian structure such that geodesics always pass through the support of data probability densities. In this section we will focus on constructing Riemannian geometry that does just that from unimodal densities $p : \mathbb{R}^d \to \mathbb{R}$ of the form

$$p(\mathbf{x}) \propto e^{-\psi(\varphi(\mathbf{x}))} \tag{3}$$

where $\psi : \mathbb{R}^d \to \mathbb{R}$ is a smooth strongly convex function and $\varphi : \mathbb{R}^d \to \mathbb{R}^d$ is a diffeomorphism, e.g., such as the density in Figure 2[3]. In particular, we will consider pullback Riemannian structures of the form

$$(\Xi, \Phi)_{\mathbf{x}}^{\nabla \psi \circ \varphi} := (D_{\mathbf{x}} \nabla \psi \circ \varphi[\Xi], D_{\mathbf{x}} \nabla \psi \circ \varphi[\Phi])_2, \tag{4}$$

which are related to the Riemannian structure obtained from the *score function* $\nabla \log(p(\cdot)) : \mathbb{R}^d \to \mathbb{R}^d$ if $\varphi$ is close to a linear $\ell^2$-isometry on the data support, i.e., $D_{\mathbf{x}}\varphi$ is an orthogonal operator:

$$(D_{\mathbf{x}} \nabla \log(p(\cdot))[\Xi], D_{\mathbf{x}} \nabla \log(p(\cdot))[\Phi])_2 = (D_{\mathbf{x}} \nabla (\psi \circ \varphi)[\Xi], D_{\mathbf{x}} \nabla (\psi \circ \varphi)[\Phi])_2$$

$$= (D_{\mathbf{x}}((D_{(\cdot)}\varphi)^\top \circ \nabla \psi \circ \varphi)[\Xi], D_{\mathbf{x}}((D_{(\cdot)}\varphi)^\top \circ \nabla \psi \circ \varphi)[\Phi])_2$$

$$\approx (D_{\mathbf{x}} \nabla \psi \circ \varphi[\Xi], D_{\mathbf{x}} \nabla \psi \circ \varphi[\Phi])_2 = (\Xi, \Phi)_{\mathbf{x}}^{\nabla \psi \circ \varphi}. \tag{5}$$

For that reason, we call such an approach to data-driven Riemannian geometry: *score-based pullback Riemannian geometry*. Since we find ourselves in a pullback setting[4], this allows to construct pullback geometry with closed-form manifold mappings.

What remains to be shown is that such geodesics and other manifold mappings pass through the data support (like in Figures 2a to 2d). The following result, which is a direct application of (Diepeveen, 2024, Prop. 2.1) and (Diepeveen, 2024, Cor. 3.6.1), gives us closed-form expressions of several important manifold mappings under $(\cdot, \cdot)^{\nabla \psi \circ \varphi}$ and makes a connection with $(\cdot, \cdot)^{\varphi}$ if we choose

$$\psi(\mathbf{x}) = \frac{1}{2}\mathbf{x}^\top \mathbf{A}^{-1}\mathbf{x}, \tag{6}$$

where $\mathbf{A} \in \mathbb{R}^{d \times d}$ is symmetric positive definite.

This special case highlights why, in general, we expect to obtain geodesics and manifold mappings that pass through the data support. For instance, in the scenario depicted in Figure 2b, where the correct form (3) is used, geodesics are computed by first reversing the effect of the diffeomorphism – transforming the data distribution to resemble a Gaussian, then drawing straight lines between the morphed data points, and finally applying the diffeomorphism again. This approach results in geodesics that traverse regions of higher likelihood between the endpoints, due to the strong convexity of the quadratic function, which aligns perfectly with our objectives.

For the proof of the result below and a more general statement and proof related to geodesics passing through the data support as in explanation above, we refer the reader to Appendix A.

**Proposition 1.** *Let $\varphi : \mathbb{R}^d \to \mathbb{R}^d$ be a smooth diffeomorphism and let $\psi : \mathbb{R}^d \to \mathbb{R}$ be a smooth strongly convex function, whose Fenchel conjugate is denoted by $\psi^\star : \mathbb{R}^d \to \mathbb{R}$. Next, consider the $\ell^2$-pullback manifolds $(\mathbb{R}^d, (\cdot, \cdot)^{\nabla \psi \circ \varphi})$ and $(\mathbb{R}^d, (\cdot, \cdot)^{\varphi})$ defined through metric tensor fields*

$$(\Xi, \Phi)_{\mathbf{x}}^{\nabla \psi \circ \varphi} := (D_{\mathbf{x}} \nabla \psi \circ \varphi[\Xi], D_{\mathbf{x}} \nabla \psi \circ \varphi[\Phi])_2, \quad and \quad (\Xi, \Phi)_{\mathbf{x}}^{\varphi} := (D_{\mathbf{x}}\varphi[\Xi], D_{\mathbf{x}}\varphi[\Phi])_2. \tag{7}$$

*Then,*

*(i) length-minimising geodesics $\gamma_{\mathbf{x},\mathbf{y}}^{\nabla \psi \circ \varphi} : [0,1] \to \mathbb{R}^d$ on $(\mathbb{R}^d, (\cdot, \cdot)^{\nabla \psi \circ \varphi})$ are given by*

$$\gamma_{\mathbf{x},\mathbf{y}}^{\nabla \psi \circ \varphi}(t) = (\varphi^{-1} \circ \nabla \psi^\star)((1-t)(\nabla \psi \circ \varphi)(\mathbf{x}) + t(\nabla \psi \circ \varphi)(\mathbf{y})). \tag{8}$$

*In addition, if $\psi$ is of the form (6)*

$$\gamma_{\mathbf{x},\mathbf{y}}^{\nabla \psi \circ \varphi}(t) = \gamma_{\mathbf{x},\mathbf{y}}^{\varphi}(t) = \varphi^{-1}((1-t)\varphi(\mathbf{x}) + t\varphi(\mathbf{y})). \tag{9}$$

---

[3]Here, $\psi(\mathbf{x}) := 2\mathbf{x}_1^2 + \frac{1}{8}\mathbf{x}_2^2$ and $\varphi(\mathbf{x}) := (\mathbf{x}_1 - \frac{1}{9}\mathbf{x}_2^2, \mathbf{x}_2)$.

[4]This is generally not true when using the score itself for probability densities of the form (3).

*(ii) the logarithmic map* $\log_{\mathbf{x}}^{\nabla\psi\circ\varphi}(\cdot) : \mathbb{R}^d \to \mathcal{T}_{\mathbf{x}}\mathbb{R}^d$ *on* $(\mathbb{R}^d, (\cdot, \cdot)^{\nabla\psi\circ\varphi})$ *is given by*

$$\log_{\mathbf{x}}^{\nabla\psi\circ\varphi} \mathbf{y} = D_{\varphi(\mathbf{x})}\varphi^{-1}[D_{(\nabla\psi\circ\varphi)(\mathbf{x})}\nabla\psi^{\star}[(\nabla\psi\circ\varphi)(\mathbf{y}) - (\nabla\psi\circ\varphi)(\mathbf{x})]]. \tag{10}$$

*In addition, if* $\psi$ *is of the form* (6)

$$\log_{\mathbf{x}}^{\nabla\psi\circ\varphi} \mathbf{y} = \log_{\mathbf{x}}^{\varphi} \mathbf{y} = D_{\varphi(\mathbf{x})}\varphi^{-1}[\varphi(\mathbf{y}) - \varphi(\mathbf{x})]. \tag{11}$$

*(iii) the exponential map* $\exp_{\mathbf{x}}^{\nabla\psi\circ\varphi}(\cdot) : \mathcal{T}_{\mathbf{x}}\mathbb{R}^d \to \mathbb{R}^d$ *on* $(\mathbb{R}^d, (\cdot, \cdot)^{\nabla\psi\circ\varphi})$ *is given by*

$$\exp_{\mathbf{x}}^{\nabla\psi\circ\varphi}(\Xi_{\mathbf{x}}) = (\varphi^{-1}\circ\nabla\psi^{\star})((\nabla\psi\circ\varphi)(\mathbf{x}) + D_{\varphi(\mathbf{x})}\nabla\psi[D_{\mathbf{x}}\varphi[\Xi_{\mathbf{x}}]]). \tag{12}$$

*In addition, if* $\psi$ *is of the form* (6)

$$\exp_{\mathbf{x}}^{\nabla\psi\circ\varphi}(\Xi_{\mathbf{x}}) = \exp_{\mathbf{x}}^{\varphi}(\Xi_{\mathbf{x}}) = \varphi^{-1}(\varphi(\mathbf{x}) + D_{\mathbf{x}}\varphi[\Xi_{\mathbf{x}}]). \tag{13}$$

*(iv) the distance* $d_{\mathbb{R}^d}^{\nabla\psi\circ\varphi} : \mathbb{R}^d \times \mathbb{R}^d \to \mathbb{R}$ *on* $(\mathbb{R}^d, (\cdot, \cdot)^{\nabla\psi\circ\varphi})$ *is given by*

$$d_{\mathbb{R}^d}^{\nabla\psi\circ\varphi}(\mathbf{x}, \mathbf{y}) = \|(\nabla\psi\circ\varphi)(\mathbf{x}) - (\nabla\psi\circ\varphi)(\mathbf{y})\|_2. \tag{14}$$

*In addition, if* $\psi$ *is of the form* (6)

$$d_{\mathbb{R}^d}^{\nabla\psi\circ\varphi}(\mathbf{x}, \mathbf{y}) = \|\varphi(\mathbf{x}) - \varphi(\mathbf{y})\|_{\mathbf{A}^{-2}} := \|\mathbf{A}^{-1}(\varphi(\mathbf{x}) - \varphi(\mathbf{y}))\|_2. \tag{15}$$

*(v) the Riemannian barycentre* $\mathbf{x}^* \in \mathbb{R}^d$ *of the data set* $\{\mathbf{x}^i\}_{i=1}^N$ *on* $(\mathbb{R}^d, (\cdot, \cdot)^{\nabla\psi\circ\varphi})$ *is given by*

$$\mathbf{x}^* := \operatorname*{arg\,min}_{\mathbf{x}\in\mathbb{R}^d}\Big\{\frac{1}{2N}\sum_{i=1}^N d_{\mathbb{R}^d}^{\nabla\psi\circ\varphi}(\mathbf{x}, \mathbf{x}^i)^2\Big\} = (\varphi^{-1}\circ\nabla\psi^{\star})\Big(\frac{1}{N}\sum_{i=1}^N \nabla\psi(\varphi(\mathbf{x}^i))\Big). \tag{16}$$

*In addition, if* $\psi$ *is of the form* (6)

$$\mathbf{x}^* := \operatorname*{arg\,min}_{\mathbf{x}\in\mathbb{R}^d}\Big\{\frac{1}{2N}\sum_{i=1}^N d_{\mathbb{R}^d}^{\varphi}(\mathbf{x}, \mathbf{x}^i)^2\Big\} = \varphi^{-1}\Big(\frac{1}{N}\sum_{i=1}^N \varphi(\mathbf{x}^i)\Big). \tag{17}$$

**Remark 1.** *We note that* $\ell^2$*-stability of geodesics and the barycentre are inherited by (Diepeveen, 2024, Thms. 3.4&3.8), if we have (approximate) local* $\ell^2$*-isometry of* $\varphi$ *on the data distribution.*

## 4 RIEMANNIAN AUTOENCODER FROM UNIMODAL PROBABILITY DENSITIES

The connection between $(\cdot, \cdot)^{\nabla\psi\circ\varphi}$ and $(\cdot, \cdot)^{\varphi}$ begs the question what $\psi$ could still be used for if it is of the form (6). We note that this case comes down to having a data probability density that is a deformed Gaussian distribution. In the case of a regular (non-deformed) Gaussian, one can compress the data generated by it through projecting them onto a low rank approximation of the covariance matrix such that only the directions with highest variance are taken into account. This is the basic idea behind PCA. In the following we will generalize this idea to the Riemannian setting and observe that this amounts to constructing a *Riemannian autoencoder* (RAE) (Diepeveen, 2024), whose error we can bound by picking the dimension of the autoencoder in a clever way, reminiscent of the classical PCA error bound.

Concretely, we assume that we have a unimodal density of the form (3) with a quadratic strongly convex function $\psi(\mathbf{x}) := \frac{1}{2}\mathbf{x}^\top\mathbf{A}^{-1}\mathbf{x}$ for some diagonal matrix $\mathbf{A} := \operatorname{diag}(\mathbf{a}_1, \dots \mathbf{a}_d)$ with positive entries[5]. Next, we define an indexing $u_w \in [d] := \{1, \dots, d\}$ for $w = 1, \dots, d$ such that

$$\mathbf{a}_{u_1} \geq \dots \geq \mathbf{a}_{u_d}, \tag{18}$$

and consider a threshold $\varepsilon \in [0, 1]$. We then consider $d_\varepsilon \in [d]$ defined as the integer that satisfies

$$d_\varepsilon := \begin{cases} \min\Big\{d' \in [d-1] \ \Big| \ \sum_{w=d'+1}^d \mathbf{a}_{u_w} \leq \varepsilon\sum_{u=1}^d \mathbf{a}_u\Big\}, & \text{if } \mathbf{a}_{u_d} \leq \varepsilon\sum_{u=1}^d \mathbf{a}_u, \\ d, & \text{otherwise.} \end{cases} \tag{19}$$

---

[5]Note that this is not restrictive as for a general symmetric positive definite matrix $\mathbf{A}$ the eigenvalues can be used as diagonal entries and the orthonormal matrices can be concatenated with the diffeomorphism.

Finally, we define the mapping $E_\varepsilon : \mathbb{R}^d \to \mathbb{R}^{d_\varepsilon}$ coordinate-wise as

$$E_\varepsilon(\mathbf{x})_w := (\log^\varphi_{\varphi^{-1}(\mathbf{0})} \mathbf{x}, D_\mathbf{0}\varphi^{-1}[\mathbf{e}^{u_w}])^\varphi_{\varphi^{-1}(\mathbf{0})} \overset{(11)}{=} (\varphi(\mathbf{x}), \mathbf{e}^{u_w})_2, \quad w = 1, \ldots, d_\varepsilon, \quad (20)$$

and define $D_\varepsilon : \mathbb{R}^{d_\varepsilon} \to \mathbb{R}^d$ as

$$D_\varepsilon(\mathbf{p}) := \exp^\varphi_{\varphi^{-1}(\mathbf{0})}\Big(\sum_{w=1}^{d_\varepsilon} \mathbf{p}_w D_\mathbf{0}\varphi^{-1}[\mathbf{e}^{u_w}]\Big) \overset{(13)}{=} \varphi^{-1}\Big(\sum_{w=1}^{d_\varepsilon} \mathbf{p}_w \mathbf{e}^{u_w}\Big), \quad (21)$$

which generate a Riemannian autoencoder and the set $D_\varepsilon(\mathbb{R}^{d_\varepsilon}) \subset \mathbb{R}^d$ as an approximate data manifold as in the scenario in Figure 1.

As hinted above, this Riemannian autoencoder comes with an error bound on the expected approximation error, which is fully determined by the diffeomorphism's deviation from isometry around the data manifold. For the proof, we refer the reader to Appendix B.

**Theorem 1.** *Let $\varphi : \mathbb{R}^d \to \mathbb{R}^d$ be a smooth diffeomorphism and let $\psi : \mathbb{R}^d \to \mathbb{R}$ be a quadratic function of the form* (6) *with positive definite diagonal matrix $\mathbf{A} \in \mathbb{R}^{d \times d}$. Furthermore, let $p : \mathbb{R}^d \to \mathbb{R}$ be the corresponding probability density of the form* (3). *Finally, consider $\varepsilon \in [0, 1]$ and the mappings $E_\varepsilon : \mathbb{R}^d \to \mathbb{R}^{d_\varepsilon}$ and $D_\varepsilon : \mathbb{R}^{d_\varepsilon} \to \mathbb{R}^d$ in* (20) *and* (21) *with $d_\varepsilon \in [d]$ as in* (19).

*Then,*

$$\mathbb{E}_{\mathbf{X} \sim p}[\|D_\varepsilon(E_\varepsilon(\mathbf{X})) - \mathbf{X}\|_2^2] \leq \varepsilon \inf_{\beta \in [0, \frac{1}{2})} \left\{ \frac{C^1_{\beta,\varphi} C^2_{\beta,\varphi} C^3_{\beta,\varphi}}{1 - 2\beta} \left(\frac{1 + \beta}{1 - 2\beta}\right)^{\frac{d}{2}} \right\} \sum_{i=1}^d \mathbf{a}_i + o(\varepsilon), \quad (22)$$

*where*

$$C^1_{\beta,\varphi} := \sup_{\mathbf{x} \in \mathbb{R}^d} \{\|D_{\varphi(\mathbf{x})}\varphi^{-1}\|_2^2 e^{-\frac{\beta}{2}\varphi(\mathbf{x})^\top \mathbf{A}^{-1}\varphi(\mathbf{x})}\}, \quad (23)$$

$$C^2_{\beta,\varphi} := \sup_{\mathbf{x} \in \mathbb{R}^d} \{|\det(D_\mathbf{x}\varphi)| e^{-\frac{\beta}{2}\varphi(\mathbf{x})^\top \mathbf{A}^{-1}\varphi(\mathbf{x})}\}, \quad (24)$$

*and*

$$C^3_{\beta,\varphi} := \sup_{\mathbf{x} \in \mathbb{R}^d} \{|\det(D_{\varphi(\mathbf{x})}\varphi^{-1})| e^{-\frac{\beta}{2}\varphi(\mathbf{x})^\top \mathbf{A}^{-1}\varphi(\mathbf{x})}\}. \quad (25)$$

**Remark 2.** *Note that the RAE latent space is interpretable as it is $\ell^2$-isometric to the data manifold if $\varphi$ is an approximate $\ell^2$-isometry on the data manifold. In other words, latent representations being close by or far away correspond to similar behaviour in data space, which is not the case for a VAE (Kingma & Welling, 2013).*

## 5 LEARNING UNIMODAL PROBABILITY DENSITIES

Naturally, we want to learn probability densities of the form (3), which can then directly be inserted into the proposed score-based pullback Riemannian geometry framework. In this section we will consider how to adapt normalizing flow (NF) (Dinh et al., 2017) training to a setting that is more suitable for our purposes[6]. In particular, we will consider how training a normalizing flow density $p : \mathbb{R}^d \to \mathbb{R}$ given by

$$p(\mathbf{x}) := \frac{1}{C_\psi} e^{-\psi(\varphi(\mathbf{x}))}|\det(D_\mathbf{x}\varphi)|, \quad (26)$$

where $C_\psi > 0$ is a normalisation constant that only depends on the strongly convex function $\psi$, yields our target distribution (3).

From Sections 3 and 4 we have seen that ideally the strongly convex function $\psi : \mathbb{R}^d \to \mathbb{R}$ corresponds to a Gaussian with a parameterised diagonal covariance matrix $\mathbf{A} \in \mathbb{R}^{d \times d}$, resulting in more parameters than in standard normalizing flows, whereas the diffeomorphism $\varphi : \mathbb{R}^d \to \mathbb{R}^d$ is regularized to be an isometry. In particular, $\mathbf{A}$ ideally allows for learnable anisotropy rather than having a fixed isotropic identity matrix. The main reason is that through anisotropy we can construct a Riemannian autoencoder (RAE), since it is known which dimensions are most important.

---

[6]We note that the choice for adapting the normalizing flow training scheme rather than using diffusion model training schemes is due to more robust results through the former.

Moreover, the diffeomorphism should be $\ell^2$-isometric, unlike standard normalizing flows which are typically non-volume preserving, enabling stability (Remark 1) and a practically useful and interpretable RAE (Theorem 1 and remark 2). In addition, $\ell^2$-isometry (on the data support) implies volume-preservation, which means that $|\det(D_\mathbf{x}\varphi)| \approx 1$ so that (26) reduces to the target distribution (3)[7].

This leads to learning the density through minimizing the following adapted normalizing flow loss

$$\mathcal{L}(\theta_1, \theta_2) := \mathbb{E}_{\mathbf{X} \sim p_{\text{data}}} \left[ -\log p_{\theta_1, \theta_2}(\mathbf{X}) \right]$$
$$+ \lambda_{\text{vol}} \mathbb{E}_{\mathbf{X} \sim p_{\text{data}}} \left[ \log(|\det(D_\mathbf{X}\varphi_{\theta_2})|)^2 \right] + \lambda_{\text{iso}} \mathbb{E}_{\mathbf{X} \sim p_{\text{data}}} \left[ \|(D_\mathbf{X}\varphi_{\theta_2})^\top D_\mathbf{X}\varphi_{\theta_2} - \mathbf{I}_d\|_F^2 \right] \quad (27)$$

where $\lambda_{\text{vol}}, \lambda_{\text{iso}} > 0$ and the negative log likelihood term reduces to

$$\mathbb{E}_{\mathbf{X} \sim p_{\text{data}}} \left[ -\log p_{\theta_1, \theta_2}(\mathbf{X}) \right] = \frac{1}{2} \mathbb{E}_{\mathbf{X} \sim p_{\text{data}}} \left[ \varphi_{\theta_2}(\mathbf{X})^\top \mathbf{A}_{\theta_1}^{-1} \varphi_{\theta_2}(\mathbf{X}) \right]$$
$$- \mathbb{E}_{\mathbf{X} \sim p_{\text{data}}} \left[ \log(|\det(D_\mathbf{X}\varphi_{\theta_2})|) \right] + \frac{1}{2} \text{tr}(\mathbf{A}_{\theta_1}) + \frac{d}{2} \log(2\pi), \quad (28)$$

where $\mathbf{A}_{\theta_1}$ is a diagonal matrix and $\varphi_{\theta_2}$ is a normalizing flow with affine coupling layers[8] (Dinh et al., 2017).

## 6 EXPERIMENTS

We conducted two sets of experiments to evaluate the proposed scheme from Section 5 to learn suitable pullback Riemannian geometry. The first set investigates whether our adaptation of the standard normalizing flow (NF) training paradigm leads to more accurate and stable manifold mappings, as measured by the geodesic and variation errors. The second set assesses the capability of our method to generate a robust Riemannian autoencoder (RAE).

For all experiments in this section, detailed training configurations are provided in Appendix E.

### 6.1 MANIFOLD MAPPINGS

As discussed in Diepeveen (2024), the quality of learned manifold mappings is determined by two key metrics: the *geodesic error* and the *variation error*. The geodesic error measures the average deviation form the ground truth geodesics implied by the ground truth pullback metric, while the variation error evaluates the stability of geodesics under small perturbations. We define these error metrics for the evaluation of pullback geometries in Appendix D.

Our approach introduces two key modifications to the normalizing flow (NF) training framework:

1. **Anisotropic Base Distribution**: We parameterize the diagonal elements of the covariance matrix $\mathbf{A}_{\theta_1}$, introducing anisotropy into the base distribution.

2. $\ell^2$**-Isometry Regularization**: We regularize the flow $\varphi_{\theta_2}$ to be approximately $\ell^2$-isometric.

To assess the effectiveness of these modifications in learning more accurate and robust manifold mappings, we compare our method against three baselines:

(1) *Normalizing Flow (NF)*: Uses an NF with a standard isotropic Gaussian base distribution $\mathcal{N}(\mathbf{0}, \mathbf{I}_d)$ and no isometry regularization of the flow.

---

[7]We note that without these constraints (accommodating multimodality) the learned mappings can in principle be used to construct Riemannian geometry and a RAE. However, from the theory discussed in this paper we cannot guarantee stability of manifold mappings nor that the RAE has the right dimension.

[8]We note that the choice for affine coupling layers rather than using more expressive diffeomorphisms such as rational quadratic flows Durkan et al. (2019) is due to our need for high regularity for stable manifold mappings (Remark 1) and an interpretable RAE (Remark 2), which has empirically shown to be more challenging to achieve for more expressive flows as both first-and higher-order derivatives of $\varphi$ will blow up the error terms in theorem 1. For more details refer to appendix G.

| Metric | Our Method | NF | Anisotropic NF | Isometric NF |
|---|---|---|---|---|
| **Single Banana Dataset** | | | | |
| Geodesic Error | **0.0315 (0.0268)** | 0.0406 (0.0288) | 0.0431 (0.0305) | 0.0817 (0.1063) |
| Variation Error | **0.0625 (0.0337)** | 0.0638 (0.0352) | 0.0639 (0.0354) | 0.0639 (0.0355) |
| **Squeezed Single Banana Dataset** | | | | |
| Geodesic Error | **0.0180 (0.0226)** | 0.0524 (0.0805) | 0.0505 (0.0787) | 0.1967 (0.2457) |
| Variation Error | **0.0631 (0.0326)** | 0.0663 (0.0353) | 0.0661 (0.0350) | 0.0669 (0.0361) |
| **River Dataset** | | | | |
| Geodesic Error | **0.1691 (0.0978)** | 0.2369 (0.1216) | 0.2561 (0.1338) | 0.3859 (0.2568) |
| Variation Error | **0.0763 (0.0486)** | 0.1064 (0.0807) | 0.1113 (0.0863) | 0.0636 (0.0333) |

Table 1: Comparison of evaluation metrics for different methods across three datasets. Best-performing results for each metric are highlighted in bold. Values are reported as mean (std). The proposed method performs best in all metrics on each data set.

(2) *Anisotropic Normalizing Flow*: Uses an NF with the same parameterization of the diagonal covariance matrix in the base distribution as in our method, but without regularization of the flow.

(3) *Isometric Normalizing Flow*: Uses an NF with an isotropic Gaussian base distribution $\mathcal{N}(\mathbf{0}, \mathbf{I}_d)$ and regularizes the flow to be approximately $\ell^2$-isometric.

We conduct experiments on three datasets, illustrated in Figure 6 in Appendix C.1: the *Single Banana Dataset*, the *Squeezed Single Banana Dataset*, and the *River Dataset*. Detailed descriptions of the construction and characteristics of these datasets are provided in Appendix C.1.

Table 1 presents the geodesic and variation errors for each method across the three datasets and Figure 3 visually compares the geodesics computed using each method on the river dataset. Our method consistently achieves significantly lower errors compared to the baselines, indicating more accurate and stable manifold mappings.

Introducing anisotropy in the base distribution without enforcing isometry in the flow offers no significant improvement over the standard flow. On the other hand, regularizing the flow to be approximately isometric without incorporating anisotropy in the base distribution results in underfitting, leading to noticeably worse performance than the standard flow. Our results demonstrate that the combination of anisotropy in the base distribution with isometry regularization (our method) yields the most accurate and stable manifold mappings, as evidenced by consistently lower geodesic and variation errors.

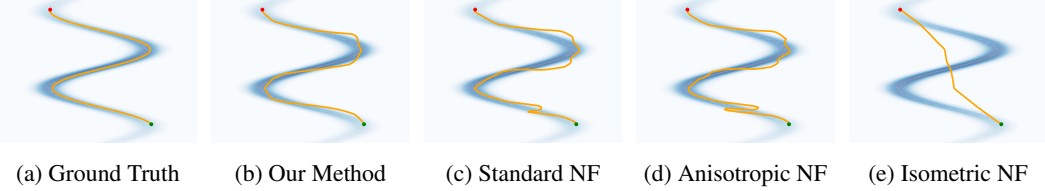

(a) Ground Truth     (b) Our Method     (c) Standard NF     (d) Anisotropic NF     (e) Isometric NF

Figure 3: Comparison of geodesics computed using different methods on the river dataset. The geodesics generated by the proposed method have least artifacts, which is in line with our expectations from Table 1.

## 6.2 RIEMANNIAN AUTOENCODER

To evaluate the capacity of our method to learn a Riemannian autoencoder, we conducted experiments on two synthetic datasets across various combinations of intrinsic dimension $d'$ and ambient dimension $d$:

- *Hemisphere*($d'$, $d$): Samples are drawn from the upper hemisphere of a $d'$-dimensional unit sphere and embedded in an $d$-dimensional ambient space via a random isometric mapping.

- *Sinusoid*($d'$, $d$): This dataset is generated by applying sinusoidal transformations to $d'$-dimensional latent variables, resulting in a complex, nonlinear manifold embedded in $d$ dimensions.

For a detailed description of these datasets, refer to Appendix C.2.

### 6.2.1 1D AND 2D MANIFOLDS

In Figures 1 and 4, we present the data manifold approximations by our Riemannian autoencoder for four low-dimensional manifolds.: Hemisphere(2,3), Sinusoid(1,3), Sinusoid(2,3) and Sinusoid(1,100). In appendix F, we detail the process used to create the data manifold approximations for these experiments. In our experiments, we set $\epsilon = 0.01$, which resulted in $d_\epsilon = d'$ for all cases, accurately capturing the intrinsic dimension of each manifold and producing accurate global charts.

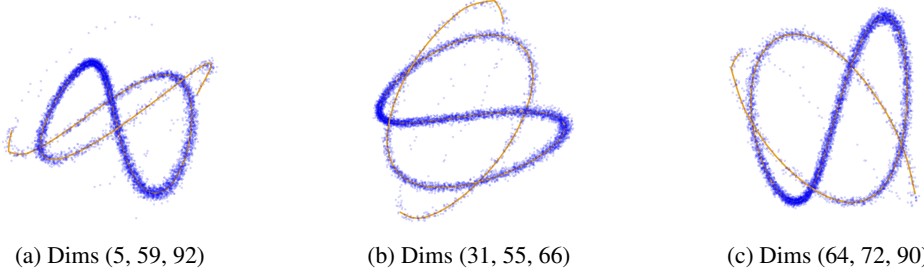

|  (a) Dims (5, 59, 92) | (b) Dims (31, 55, 66) | (c) Dims (64, 72, 90) |

Figure 4: Approximate data manifold learned by the Riemannian autoencoder for the Sinusoid(1, 100) dataset. The orange curves depict the manifold learned by the model, while the blue points show the training data. We visualize three different combinations of the ambient dimensions.

### 6.2.2 HIGHER-DIMENSIONAL MANIFOLDS

To evaluate the scalability of our method to higher-dimensional manifolds, we conducted additional experiments on the Hemisphere(5,20) and Sinusoid(5,20) datasets.

Our theory suggests that the learned variances indicate the importance of each latent dimension: higher variances signal more important dimensions for reconstructing the manifold, while dimensions with vanishing variances are considered insignificant and are disregarded when constructing the Riemannian autoencoder. To test the model's ability to correctly identify important and unimportant latent dimensions, we report the average $\ell^2$ reconstruction error for each dataset as a function of the number of latent dimensions used. In the reconstruction error plots (see figs. 5b and 5d), we report three variance-based orders for adding latent dimensions: decreasing variance order (blue line), increasing variance order (green line), and random order (red line).

For the Hemisphere(5,20) dataset, the model identified five non-vanishing variances (see fig. 5a), perfectly capturing the intrinsic dimension of the manifold. This is reflected in the blue curve in fig. 5b, where the first five latent dimensions, corresponding to the largest variances, are sufficient to reduce the reconstruction error almost to zero. In contrast, the green curve illustrates that the remaining ambient dimensions do not encode useful information about the manifold. The red curve demonstrates improvement only when an important latent dimension is included.

For the more challenging Sinusoid(5,20) dataset, our method still performs very well, though not as perfectly as for the Hemisphere dataset. The first six most important latent dimensions explain approximately $97\%$ of the variance, increasing to over $99\%$ with the seventh dimension (see fig. 5c). This is reflected in the blue curve in fig. 5d, where the first six latent dimensions reduce the reconstruction error to near zero, and the addition of the seventh dimension brings the error effectively to zero. The slight discrepancy between our results and the ground truth likely arises from increased optimization difficulty, as the normalizing flow must learn a more intricate distribution while maintaining approximate isometry. We believe that with deeper architectures and more careful tuning of the optimization loss, the model will converge to the correct intrinsic dimensionality of five. Currently, it predicts six dimensions at a threshold of $\epsilon = 0.05$ and seven at $\epsilon = 0.01$, slightly overestimating due to the manifold's complexity.

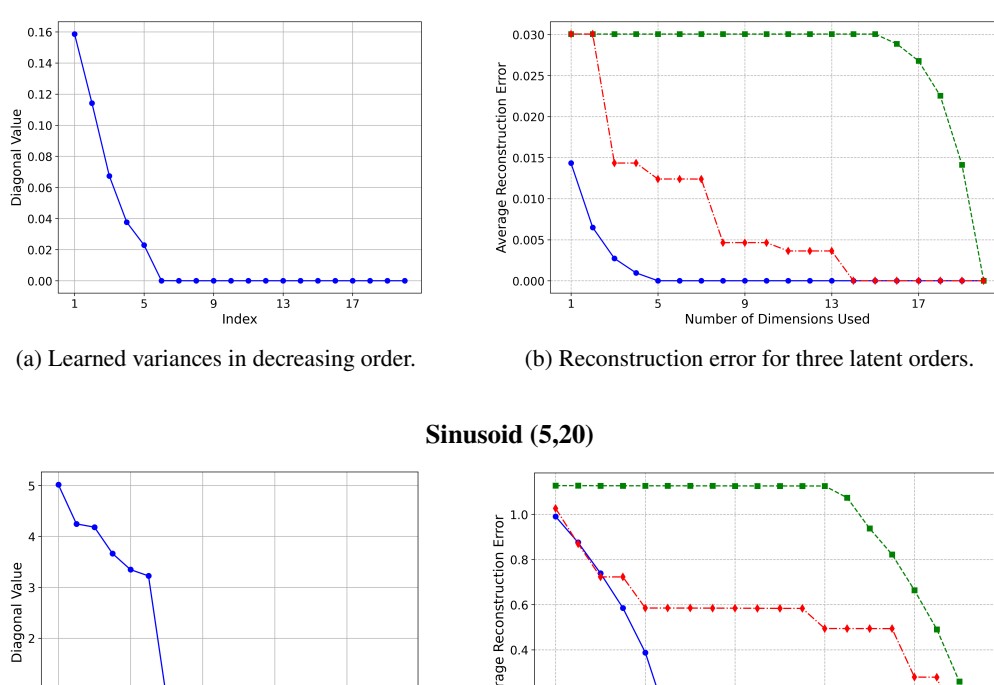

Figure 5: Learned variances and reconstruction errors for the Hemisphere(5,20) and Sinusoid(5,20) datasets. The plots in the left column show the learned variances in decreasing order for each dataset, while the right column illustrates the average $\ell^2$ reconstruction error as a function of the number of latent dimensions used. The reconstruction errors are evaluated for three variance-based orders of the latent dimensions: the **blue line** (circular markers) represents adding dimensions in decreasing order of variance, the **green line** (square markers) for increasing variance, and the **red line** (diamond markers) for a random order.

## 7 CONCLUSIONS

In this work we have taken a first step towards a practical data-driven Riemannian geometry framework, striking a balance between scalability of training a data-driven Riemannian structure and of evaluating its corresponding manifold mappings. We have considered a family of unimodal probability densities whose negative log-likelihoods are compositions of strongly convex functions and diffeomorphisms, and sought to learn them. We have shown that once these unimodal densities have been learned, the proposed score-based pullback geometry gives us closed-form geodesics that pass through the data probability density and a Riemannian autoencoder with error bounds that can be used to estimate the dimension of the data manifold. Finally, to learn the distribution we have proposed an adaptation to normalizing flow training. Through numerical experiments, we have shown that these modifications are crucial for extracting geometric information, and that our framework not only generates high-quality geodesics across the data support, but also accurately estimates the intrinsic dimension of the approximate data manifold while constructing a global chart, even in high-dimensional ambient spaces. Current challenges of the method lie in balancing the expressivity of the network architecture, e.g., through additional layers or more expressive architectures, and satisfying approximate $\ell^2$-isometry on the data support. For future work we aim to overcome these challenges, extending the method to multimodal distributions, while making it scalable for higher-dimensional data sets. After that, we believe that this line of work has wide variety of downstream applications as many of the applications mentioned to motivate this line of work will benefit from more interpretable representation learning.

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

## A    PROOF OF PROPOSITION 1 AND AN ADDITIONAL RESULT

*Proof of proposition 1.* First note that $\nabla\psi \circ \varphi$ is a diffeomorphism with inverse $\varphi^{-1} \circ \nabla\psi^\star$. Then, equations (8), (10), (12), and (14) follow directly from (Diepeveen, 2024, Prop. 2.1) and (16) follows directly from (Diepeveen, 2024, Cor. 3.6.1).

Next, if $\psi$ is of the form (6), i.e.,

$$\psi(\mathbf{x}) = \frac{1}{2}\mathbf{x}^\top \mathbf{A}^{-1}\mathbf{x},$$

we have that its Fenchel conjugate is given by

$$\psi^\star(\mathbf{y}) = \frac{1}{2}\mathbf{y}^\top \mathbf{A}\mathbf{y}. \tag{29}$$

So both $\nabla\psi(\mathbf{x}) = \mathbf{A}^{-1}\mathbf{x}$ and $\nabla\psi^\star(\mathbf{y}) = \mathbf{A}\mathbf{y}$ are linear mappings, from which follows that they cancel to identity everywhere and yield (9), (11), (13), (15), and (17). □

**Proposition 2.** *Let $\varphi : \mathbb{R}^d \to \mathbb{R}^d$ be a smooth diffeomorphism and let $\psi : \mathbb{R}^d \to \mathbb{R}$ be a smooth strongly convex function, whose Fenchel conjugate is denoted by $\psi^\star : \mathbb{R}^d \to \mathbb{R}$. Next, consider the function $f : \mathbb{R}^d \to \mathbb{R}^{d \times d}$ given by*

$$f(\mathbf{z}) := D_{\mathbf{z}}\nabla\psi^\star + \sum_{i=1}^{d} \mathbf{z}_i \partial_i D_{(\cdot)}\nabla\psi^\star. \tag{30}$$

*Finally, let $\mathbf{x}, \mathbf{y} \in \mathbb{R}^d$ be vectors and assume that for all vectors*

$$\mathbf{z} \in \{(1-t)(\nabla\psi \circ \varphi)(\mathbf{x}) + t(\nabla\psi \circ \varphi)(\mathbf{y}) \mid t \in [0,1]\} \subset \mathbb{R}^d$$

*the matrix $f(\mathbf{z})$ is positive definite.*

*Then, mapping*

$$t \mapsto \psi(\varphi(\gamma_{\mathbf{x},\mathbf{y}}^{\nabla\psi\circ\varphi}(t))), \quad t \in [0,1] \tag{31}$$

*is strongly convex, where $\gamma_{\mathbf{x},\mathbf{y}}^{\nabla\psi\circ\varphi}$ is the geodesic between $\mathbf{x}$ and $\mathbf{y}$ under the Riemannian structure $(\mathbb{R}^d, (\cdot, \cdot)^{\nabla\psi\varphi})$.*

*In addition, if $\psi$ is of the form (6) the mapping (31) is strongly convex for any $\mathbf{x}, \mathbf{y} \in \mathbb{R}^d$.*

*Proof.* By (8) in proposition 1 we have

$$\psi(\varphi(\gamma_{\mathbf{x},\mathbf{y}}^{\nabla\psi\circ\varphi}(t))) = \psi(\varphi((\varphi^{-1} \circ \nabla\psi^\star)((1-t)(\nabla\psi\circ\varphi)(\mathbf{x}) + t(\nabla\psi\circ\varphi)(\mathbf{y}))))$$
$$= \psi(\nabla\psi^\star((1-t)(\nabla\psi\circ\varphi)(\mathbf{x}) + t(\nabla\psi\circ\varphi)(\mathbf{y}))). \tag{32}$$

So the claim holds if on the linear subspace

$$\{(1-t)(\nabla\psi\circ\varphi)(\mathbf{x}) + t(\nabla\psi\circ\varphi)(\mathbf{y}) \mid t \in [0,1]\} \subset \mathbb{R}^d \tag{33}$$

the function $\psi \circ \nabla\psi^\star$ is convex.

Next, note that the Hessian of $\psi \circ \nabla\psi^\star$ satisfies

$$D_{\mathbf{z}}\nabla(\psi \circ \nabla\psi^\star) = f(\mathbf{z}). \tag{34}$$

By assumption $f(\mathbf{z})$ is positive definite for all $\mathbf{z}$ in the subspace (33). In other words, on this subspace $\psi(\nabla\psi^\star(\mathbf{z}))$ is positive definite, which implies strong convexity and yields the main claim.

The claim for the special case of $\psi$ is of the form (6) follows directly, because

$$f(\mathbf{z}) = \mathbf{A}, \tag{35}$$

which is always positive definite.

□

## B   PROOF OF THEOREM 1

**Auxiliary lemma**

**Lemma 1.** *Let $\varphi : \mathbb{R}^d \to \mathbb{R}^d$ be a smooth diffeomorphism and let $\psi : \mathbb{R}^d \to \mathbb{R}$ be a quadratic function of the form* (6) *with diagonal $\mathbf{A} \in \mathbb{R}^{d \times d}$. Furthermore, let $p : \mathbb{R}^d \to \mathbb{R}$ be the corresponding probability density of the form* (3). *Finally, consider $\varepsilon \in [0, 1]$ and the mappings $E_\varepsilon : \mathbb{R}^d \to \mathbb{R}^{d_\varepsilon}$ and $D_\varepsilon : \mathbb{R}^{d_\varepsilon} \to \mathbb{R}^d$ in* (20) *and* (21) *with $d_\varepsilon \in [d]$ as in* (19).

*Then, for any $\alpha \in [0, 1)$ and any $\beta \in [0, 1 - \alpha)$*

$$\mathbb{E}_{\mathbf{X} \sim p}[d_{\mathbb{R}^d}^\varphi(D_\varepsilon(E_\varepsilon(\mathbf{X})), \mathbf{X})^2 e^{\frac{\alpha}{2}\varphi(\mathbf{X})^\top \mathbf{A}^{-1}\varphi(\mathbf{X})}] \le \varepsilon \frac{C_{\beta,\varphi}^2 C_{\beta,\varphi}^3}{1 - \alpha - \beta}\Big(\frac{1 + \beta}{1 - \alpha - \beta}\Big)^{\frac{d}{2}}\sum_{i=1}^d \mathbf{a}_i, \quad (36)$$

*where*

$$C_{\beta,\varphi}^3 := \sup_{\mathbf{x} \in \mathbb{R}^d}\{|\det(D_{\varphi(\mathbf{x})}\varphi^{-1})|e^{-\frac{\beta}{2}\varphi(\mathbf{x})^\top \mathbf{A}^{-1}\varphi(\mathbf{x})}\}, \quad (37)$$

*and*

$$C_{\beta,\varphi}^2 := \sup_{\mathbf{x} \in \mathbb{R}^d}\{|\det(D_{\mathbf{x}}\varphi)|e^{-\frac{\beta}{2}\varphi(\mathbf{x})^\top \mathbf{A}^{-1}\varphi(\mathbf{x})}\}. \quad (38)$$

*Proof.* We need to distinct two cases: (i) $d_\varepsilon = d$ and (ii) $1 \le d_\varepsilon < d$

(i) If $d_\varepsilon = d$ we have that $D_\varepsilon(E_\varepsilon(\mathbf{x})) = \mathbf{x}$ for any $\mathbf{x} \in \mathbb{R}^d$. In other words

$$\mathbb{E}_{\mathbf{X} \sim p}[d_{\mathbb{R}^d}^\varphi(D_\varepsilon(E_\varepsilon(\mathbf{X})), \mathbf{X})^2 e^{\frac{\alpha}{2}\varphi(\mathbf{X})^\top \mathbf{A}^{-1}\varphi(\mathbf{X})}] = 0 \le \varepsilon \frac{C_{\beta,\varphi}^2 C_{\beta,\varphi}^3}{1 - \alpha - \beta}\Big(\frac{1 + \beta}{1 - \alpha - \beta}\Big)^{\frac{d}{2}}\sum_{i=1}^d \mathbf{a}_i. \quad (39)$$

(ii) Next, we consider the case $1 \le d_\varepsilon < d$. First, notice that we can rewrite

$$\|\varphi(D_\varepsilon(E_\varepsilon(\mathbf{x}))) - \varphi(\mathbf{x})\|_2^2 \stackrel{\text{(20) and (21)}}{=} \|\sum_{k=1}^{d_\varepsilon}(\varphi(\mathbf{x}), \mathbf{e}^{i_k})_2\mathbf{e}^{i_k} - \varphi(\mathbf{x})\|_2^2 = \|\sum_{k=d_\varepsilon+1}^d (\varphi(\mathbf{x}), \mathbf{e}^{i_k})_2\mathbf{e}^{i_k}\|_2^2$$

$$\stackrel{\text{orthogonality}}{=} \sum_{k=d_\varepsilon+1}^d \|(\varphi(\mathbf{x}), \mathbf{e}^{i_k})_2\mathbf{e}^{i_k}\|_2^2 = \sum_{k=d_\varepsilon+1}^d (\varphi(\mathbf{x}), \mathbf{e}^{i_k})_2^2 = \sum_{k=d_\varepsilon+1}^d \varphi(\mathbf{x})_{i_k}^2. \quad (40)$$

Moreover, we define

$$C := \int_{\mathbb{R}^d} e^{-\frac{1}{2}\varphi(\mathbf{x})^\top \mathbf{A}^{-1}\varphi(\mathbf{x})}\mathrm{d}\mathbf{x}. \quad (41)$$

Then,

$$\mathbb{E}_{\mathbf{X} \sim p}[d_{\mathbb{R}^d}^\varphi(D_\varepsilon(E_\varepsilon(\mathbf{X})), \mathbf{X})^2 e^{\frac{\alpha}{2}\varphi(\mathbf{X})^\top \mathbf{A}^{-1}\varphi(\mathbf{X})}] = \frac{\int_{\mathbb{R}^d}\|\varphi(D_\varepsilon(E_\varepsilon(\mathbf{x}))) - \varphi(\mathbf{x})\|_2^2 e^{-(\frac{1}{2} - \frac{\alpha}{2})\varphi(\mathbf{x})^\top \mathbf{A}^{-1}\varphi(\mathbf{x})}\mathrm{d}\mathbf{x}}{\int_{\mathbb{R}^d} e^{-\frac{1}{2}\varphi(\mathbf{x})^\top \mathbf{A}^{-1}\varphi(\mathbf{x})}\mathrm{d}\mathbf{x}}$$

$$\stackrel{\text{(41)}}{=} \frac{1}{C}\int_{\mathbb{R}^d}\|\varphi(D_\varepsilon(E_\varepsilon(\mathbf{x}))) - \varphi(\mathbf{x})\|_2^2 e^{-(\frac{1}{2} - \frac{\alpha}{2})\varphi(\mathbf{x})^\top \mathbf{A}^{-1}\varphi(\mathbf{x})}\mathrm{d}\mathbf{x}$$

$$\stackrel{\text{(40)}}{=} \frac{1}{C}\int_{\mathbb{R}^d}\sum_{k=d_\varepsilon+1}^d \varphi(\mathbf{x})_{i_k}^2 e^{-(\frac{1}{2} - \frac{\alpha}{2})\varphi(\mathbf{x})^\top \mathbf{A}^{-1}\varphi(\mathbf{x})}\mathrm{d}\mathbf{x} = \frac{1}{C}\sum_{k=d_\varepsilon+1}^d \int_{\mathbb{R}^d}\varphi(\mathbf{x})_{i_k}^2 e^{-(\frac{1}{2} - \frac{\alpha}{2})\varphi(\mathbf{x})^\top \mathbf{A}^{-1}\varphi(\mathbf{x})}\mathrm{d}\mathbf{x}$$

$$\stackrel{\mathbf{x}=\varphi^{-1}(\mathbf{y})}{=} \frac{1}{C}\sum_{k=d_\varepsilon+1}^d \int_{\mathbb{R}^d}\mathbf{y}_{i_k}^2 e^{-(\frac{1}{2} - \frac{\alpha}{2})\mathbf{y}^\top \mathbf{A}^{-1}\mathbf{y}}|\det(D_{\mathbf{y}}\varphi^{-1})|\mathrm{d}\mathbf{y}$$

$$= \frac{1}{C}\sum_{k=d_\varepsilon+1}^d \int_{\mathbb{R}^d}\mathbf{y}_{i_k}^2 e^{-(\frac{1}{2} - \frac{\alpha}{2} - \frac{\beta}{2})\mathbf{y}^\top \mathbf{A}^{-1}\mathbf{y}}|\det(D_{\mathbf{y}}\varphi^{-1})|e^{-\frac{\beta}{2}\mathbf{y}^\top \mathbf{A}^{-1}\mathbf{y}}\mathrm{d}\mathbf{y}$$

$$\le \frac{\sup_{\mathbf{y} \in \mathbb{R}^d}\{|\det(D_{\mathbf{y}}\varphi^{-1})|e^{-\frac{\beta}{2}\mathbf{y}^\top \mathbf{A}^{-1}\mathbf{y}}\}}{C}\sum_{k=d_\varepsilon+1}^d \int_{\mathbb{R}^d}\mathbf{y}_{i_k}^2 e^{-(\frac{1}{2} - \frac{\alpha}{2} - \frac{\beta}{2})\mathbf{y}^\top \mathbf{A}^{-1}\mathbf{y}}\mathrm{d}\mathbf{y}$$

$$\overset{(37)}{=} \frac{C_{\beta,\varphi}^2}{C} \sum_{k=d_\varepsilon+1}^{d} \int_{\mathbb{R}^d} \mathbf{y}_{i_k}^2 e^{-(\frac{1}{2}-\frac{\alpha}{2}-\frac{\beta}{2})\mathbf{y}^\top \mathbf{A}^{-1}\mathbf{y}}\mathrm{d}\mathbf{y} = \frac{C_{\beta,\varphi}^2}{C} \sum_{k=d_\varepsilon+1}^{d} \int_{\mathbb{R}^d} \mathbf{y}_{i_k}^2 e^{-(\frac{1}{2}-\frac{\alpha}{2}-\frac{\beta}{2})\sum_{j=1}^{d} \frac{\mathbf{y}_j^2}{\mathbf{a}_j}}\mathrm{d}\mathbf{y}$$

$$= \frac{C_{\beta,\varphi}^2}{C} \sum_{k=d_\varepsilon+1}^{d} \int_{\mathbb{R}} \mathbf{y}_{i_k}^2 e^{-(\frac{1}{2}-\frac{\alpha}{2}-\frac{\beta}{2})\frac{\mathbf{y}^2}{\mathbf{a}_{i_k}}}\mathrm{d}\mathbf{y}_{i_k} \int_{\mathbb{R}^{d-1}} e^{-(\frac{1}{2}-\frac{\alpha}{2}-\frac{\beta}{2})\sum_{j\neq i_k}^{d}\frac{\mathbf{y}_j^2}{\mathbf{a}_j}}\mathrm{d}\mathbf{y}_1\ldots\mathrm{d}\mathbf{y}_{i_k-1}\mathrm{d}\mathbf{y}_{i_k+1}\ldots\mathrm{d}\mathbf{y}_d$$

$$= \frac{C_{\beta,\varphi}^2}{C} \sum_{k=d_\varepsilon+1}^{d} \frac{\mathbf{a}_{i_k}}{(1-\alpha-\beta)} \int_{\mathbb{R}} e^{-(\frac{1}{2}-\frac{\alpha}{2}-\frac{\beta}{2})\frac{\mathbf{y}^2}{\mathbf{a}_{i_k}}}\mathrm{d}\mathbf{y}_{i_k} \int_{\mathbb{R}^{d-1}} e^{-(\frac{1}{2}-\frac{\alpha}{2}-\frac{\beta}{2})\sum_{j\neq i_k}^{d}\frac{\mathbf{y}_j^2}{\mathbf{a}_j}}\mathrm{d}\mathbf{y}_1\ldots\mathrm{d}\mathbf{y}_{i_k-1}\mathrm{d}\mathbf{y}_{i_k+1}\ldots\mathrm{d}\mathbf{y}_d$$

$$= \frac{C_{\beta,\varphi}^2}{C} \sum_{k=d_\varepsilon+1}^{d} \frac{\mathbf{a}_{i_k}}{(1-\alpha-\beta)} \int_{\mathbb{R}^d} e^{-(\frac{1}{2}-\frac{\alpha}{2}-\frac{\beta}{2})\mathbf{y}^\top \mathbf{A}^{-1}\mathbf{y}}\mathrm{d}\mathbf{y}$$

$$= \frac{C_{\beta,\varphi}^2}{C} \sum_{k=d_\varepsilon+1}^{d} \frac{\mathbf{a}_{i_k}}{(1-\alpha-\beta)} \Big(\frac{1+\beta}{1-\alpha-\beta}\Big)^{\frac{d}{2}} \int_{\mathbb{R}^d} e^{-(\frac{1}{2}+\frac{\beta}{2})\mathbf{y}^\top \mathbf{A}^{-1}\mathbf{y}}\mathrm{d}\mathbf{y}$$

$$\overset{\mathbf{y}=\varphi(\mathbf{x})}{=} \frac{C_{\beta,\varphi}^2}{C} \sum_{k=d_\varepsilon+1}^{d} \frac{\mathbf{a}_{i_k}}{(1-\alpha-\beta)} \Big(\frac{1+\beta}{1-\alpha-\beta}\Big)^{\frac{d}{2}} \int_{\mathbb{R}^d} e^{-(\frac{1}{2}+\frac{\beta}{2})\varphi(\mathbf{x})^\top \mathbf{A}^{-1}\varphi(\mathbf{x})}|\det(D_\mathbf{x}\varphi)|\mathrm{d}\mathbf{x}$$

$$= \frac{C_{\beta,\varphi}^2}{C} \sum_{k=d_\varepsilon+1}^{d} \frac{\mathbf{a}_{i_k}}{(1-\alpha-\beta)} \Big(\frac{1+\beta}{1-\alpha-\beta}\Big)^{\frac{d}{2}} \int_{\mathbb{R}^d} e^{-\frac{1}{2}\varphi(\mathbf{x})^\top \mathbf{A}^{-1}\varphi(\mathbf{x})}|\det(D_\mathbf{x}\varphi)|e^{-\frac{\beta}{2}\varphi(\mathbf{x})^\top \mathbf{A}^{-1}\varphi(\mathbf{x})}\mathrm{d}\mathbf{x}$$

$$\leq \frac{C_{\beta,\varphi}^2 \sup_{\mathbf{x}\in\mathbb{R}^d}\{|\det(D_\mathbf{x}\varphi)|e^{-\frac{\beta}{2}\varphi(\mathbf{x})^\top \mathbf{A}^{-1}\varphi(\mathbf{x})}\}}{C} \sum_{k=d_\varepsilon+1}^{d} \frac{\mathbf{a}_{i_k}}{(1-\alpha-\beta)} \Big(\frac{1+\beta}{1-\alpha-\beta}\Big)^{\frac{d}{2}} \int_{\mathbb{R}^d} e^{-\frac{1}{2}\varphi(\mathbf{x})^\top \mathbf{A}^{-1}\varphi(\mathbf{x})}\mathrm{d}\mathbf{x}$$

$$\overset{(38)}{=} \frac{C_{\beta,\varphi}^2 C_{\beta,\varphi}^3}{C} \sum_{k=d_\varepsilon+1}^{d} \frac{\mathbf{a}_{i_k}}{(1-\alpha-\beta)} \Big(\frac{1+\beta}{1-\alpha-\beta}\Big)^{\frac{d}{2}} \int_{\mathbb{R}^d} e^{-\frac{1}{2}\varphi(\mathbf{x})^\top \mathbf{A}^{-1}\varphi(\mathbf{x})}\mathrm{d}\mathbf{x}$$

$$\overset{(41)}{=} \frac{C_{\beta,\varphi}^2 C_{\beta,\varphi}^3}{1-\alpha-\beta} \Big(\frac{1+\beta}{1-\alpha-\beta}\Big)^{\frac{d}{2}} \sum_{k=d_\varepsilon+1}^{d} \mathbf{a}_{i_k}$$

$$\overset{(19)}{\leq} \varepsilon \frac{C_{\beta,\varphi}^2 C_{\beta,\varphi}^3}{1-\alpha-\beta} \Big(\frac{1+\beta}{1-\alpha-\beta}\Big)^{\frac{d}{2}} \sum_{i=1}^{d} \mathbf{a}_i. \quad (42)$$

$$\square$$

**Proof of the theorem**

*Proof of theorem 1.* First, consider the Taylor approximation

$$\varphi^{-1}(\varphi(\mathbf{y})) - \varphi^{-1}(\varphi(\mathbf{y})) = D_{\varphi(\mathbf{x})}\varphi^{-1}[\varphi(\mathbf{y})-\varphi(\mathbf{x})] + \mathcal{O}(\|\varphi(\mathbf{y})-\varphi(\mathbf{x})\|_2^2)$$
$$= D_{\varphi(\mathbf{x})}\varphi^{-1}[\varphi(\mathbf{y})-\varphi(\mathbf{x})] + \mathcal{O}(d_{\mathbb{R}^d}^\varphi(\mathbf{y},\mathbf{x})^2). \quad (43)$$

Moreover, we define

$$C := \int_{\mathbb{R}^d} e^{-\frac{1}{2}\varphi(\mathbf{x})^\top \mathbf{A}^{-1}\varphi(\mathbf{x})}\mathrm{d}\mathbf{x}. \quad (44)$$

Subsequently, notice that

$$\mathbb{E}_{\mathbf{X}\sim p}[\|D_{\varphi(\mathbf{X})}\varphi^{-1}[\varphi(D_\varepsilon(E_\varepsilon(\mathbf{X})))-\varphi(\mathbf{X})]\|_2^2]$$

$$= \frac{1}{C} \int_{\mathbb{R}^d} \|D_{\varphi(\mathbf{x})}\varphi^{-1}[\varphi(D_\varepsilon(E_\varepsilon(\mathbf{x})))-\varphi(\mathbf{x})]\|_2^2 e^{-\frac{1}{2}\varphi(\mathbf{x})^\top \mathbf{A}^{-1}\varphi(\mathbf{x})}\mathrm{d}\mathbf{x}$$

$$\leq \frac{1}{C} \int_{\mathbb{R}^d} \|D_{\varphi(\mathbf{x})}\varphi^{-1}\|_2^2 \|\varphi(D_\varepsilon(E_\varepsilon(\mathbf{x})))-\varphi(\mathbf{x})\|_2^2 e^{-\frac{1}{2}\varphi(\mathbf{x})^\top \mathbf{A}^{-1}\varphi(\mathbf{x})}\mathrm{d}\mathbf{x}$$

$$\leq \frac{\sup_{\mathbf{x}\in\mathbb{R}^d}\{\|D_{\varphi(\mathbf{x})}\varphi^{-1}\|_2^2 e^{-\frac{\beta}{2}\varphi(\mathbf{x})^\top \mathbf{A}^{-1}\varphi(\mathbf{x})}\}}{C} \int_{\mathbb{R}^d} \|\varphi(D_\varepsilon(E_\varepsilon(\mathbf{x})))-\varphi(\mathbf{x})\|_2^2 e^{-(\frac{1}{2}-\frac{\beta}{2})\varphi(\mathbf{x})^\top \mathbf{A}^{-1}\varphi(\mathbf{x})}\mathrm{d}\mathbf{x}$$

$$\overset{(23)}{=} \frac{C^1_{\beta,\varphi}}{C} \int_{\mathbb{R}^d} \|\varphi(D_\varepsilon(E_\varepsilon(\mathbf{x}))) - \varphi(\mathbf{x})\|_2^2 e^{\frac{\beta}{2}\varphi(\mathbf{x})^\top \mathbf{A}^{-1}\varphi(\mathbf{x})} e^{-\frac{1}{2}\varphi(\mathbf{x})^\top \mathbf{A}^{-1}\varphi(\mathbf{x})} \mathrm{d}\mathbf{x}$$

$$= C^1_{\beta,\varphi}\mathbb{E}_{\mathbf{X}\sim p}[d^\varphi_{\mathbb{R}^d}(D_\varepsilon(E_\varepsilon(\mathbf{X})), \mathbf{X})^2 e^{\frac{\beta}{2}\varphi(\mathbf{X})^\top \mathbf{A}^{-1}\varphi(\mathbf{X})}]$$

$$\overset{\text{lemma 1}}{\leq} \varepsilon \frac{C^1_{\beta,\varphi}C^2_{\beta,\varphi}C^3_{\beta,\varphi}}{1-2\beta}\Big(\frac{1+\beta}{1-2\beta}\Big)^{\frac{d}{2}}\sum_{i=1}^d \mathbf{a}_i. \quad (45)$$

Then,

$$\mathbb{E}_{\mathbf{X}\sim p}[\|D_\varepsilon(E_\varepsilon(\mathbf{X})) - \mathbf{X}\|_2^2] = \mathbb{E}_{\mathbf{X}\sim p}[\|\varphi^{-1}(\varphi(D_\varepsilon(E_\varepsilon(\mathbf{X})))) - \varphi^{-1}(\varphi(\mathbf{X}))\|_2^2]$$

$$\overset{(43)}{=} \mathbb{E}_{\mathbf{X}\sim p}[\|D_{\varphi(\mathbf{X})}\varphi^{-1}[\varphi(D_\varepsilon(E_\varepsilon(\mathbf{X}))) - \varphi(\mathbf{X})] + \mathcal{O}(d^\varphi_{\mathbb{R}^d}(D_\varepsilon(E_\varepsilon(\mathbf{X})), \mathbf{X})^2)\|_2^2]$$

$$= \mathbb{E}_{\mathbf{X}\sim p}[\|D_{\varphi(\mathbf{X})}\varphi^{-1}[\varphi(D_\varepsilon(E_\varepsilon(\mathbf{X}))) - \varphi(\mathbf{X})]\|_2^2 + \mathcal{O}(d^\varphi_{\mathbb{R}^d}(D_\varepsilon(E_\varepsilon(\mathbf{X})), \mathbf{X})^3)]$$

$$\overset{(45)}{\leq} \varepsilon \frac{C^1_{\beta,\varphi}C^2_{\beta,\varphi}C^3_{\beta,\varphi}}{1-2\beta}\Big(\frac{1+\beta}{1-2\beta}\Big)^{\frac{d}{2}}\sum_{i=1}^d \mathbf{a}_i + o(\varepsilon), \quad (46)$$

which yields the claim as $\beta$ was arbitrary. $\qquad\square$

## C  DATASET CONSTRUCTION DETAILS

In this section, we provide a detailed explanation of the construction of the datasets used in our experiments. We organize the datasets into two categories based on the experimental sections in which they are used.

### C.1  DATASETS FOR MANIFOLD MAPPING EXPERIMENTS

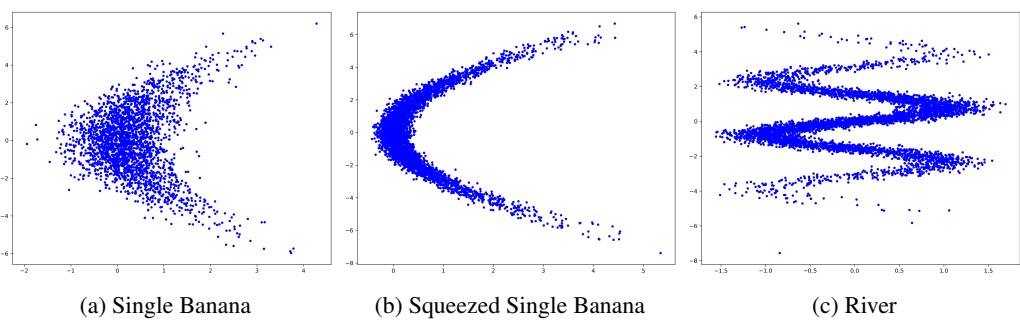

(a) Single Banana       (b) Squeezed Single Banana       (c) River

Figure 6: Visualization of the datasets used in our manifold mapping experiments.

In our manifold mapping experiments (Section 6.1), we use the following datasets illustrated in Figure 6:

- *Single Banana Dataset*: A two-dimensional dataset shaped like a curved banana.
- *Squeezed Single Banana Dataset*: A variant of the Single Banana with a tighter bend.
- *River Dataset*: A more complex 2D dataset resembling the meandering path of a river.

Each dataset is constructed by defining specific diffeomorphisms $\varphi$ and convex quadratic functions $\psi$, then sampling from the resulting probability density using Langevin Monte Carlo Markov Chain (MCMC) with Metropolis-Hastings correction. The probability density function is defined as:

$$p(\mathbf{x}) \propto e^{-\psi(\varphi(\mathbf{x}))}, \quad (47)$$

where the strongly convex function $\psi$ is given by:

$$\psi(\mathbf{v}) = \frac{1}{2}\mathbf{v}^\top A^{-1}\mathbf{v}, \tag{48}$$

and $A$ is a positive-definite diagonal matrix. The specific choices of $\varphi$ and $A$ for each dataset determine its geometric properties.

### C.1.1  DIFFEOMORPHISMS AND CONVEX QUADRATIC FUNCTIONS

The key differences between the datasets arise from the diffeomorphism $\varphi$ and the covariance matrix $\mathbf{A}$ used in the sampling process. Below, we describe the specific settings for each dataset.

**1. Single Banana Dataset**

- Diffeomorphism:

$$\varphi(\mathbf{x}) = \begin{pmatrix} x_1 - ax_2^2 - z \\ x_2 \end{pmatrix}$$

  where $a = \frac{1}{9}$ and $z = 0$.
- Covariance matrix:

$$\mathbf{A} = \begin{pmatrix} \frac{1}{4} & 0 \\ 0 & 4 \end{pmatrix}$$

**2. Squeezed Single Banana Dataset**

- Diffeomorphism: Same as the Single Banana Dataset.
- Covariance matrix:

$$\mathbf{A} = \begin{pmatrix} \frac{1}{81} & 0 \\ 0 & 4 \end{pmatrix}$$

**3. River Dataset**

- Diffeomorphism:

$$\varphi(\mathbf{x}) = \begin{pmatrix} x_1 - \sin(ax_2) - z \\ x_2 \end{pmatrix}$$

  where $a = 2$ and $z = 0$.
- Covariance matrix:

$$\mathbf{A} = \begin{pmatrix} \frac{1}{25} & 0 \\ 0 & 3 \end{pmatrix}$$

### C.1.2  DATASET GENERATION ALGORITHM

Algorithm 1 outlines the dataset generation process for all three datasets. The specific diffeomorphisms and quadratic functions differ for each dataset.

### C.2  DATASETS FOR RIEMANNIAN AUTOENCODER EXPERIMENTS

In the Riemannian autoencoder experiments (Section 6.2), we use the following datasets:

- *Hemisphere*($d'$, $d$) Dataset: Samples drawn from the upper hemisphere of a $d'$-dimensional unit sphere and embedded into $\mathbb{R}^d$ via a random isometric mapping.
- *Sinusoid*($d'$, $d$) Dataset: Generated by applying sinusoidal transformations to $d'$-dimensional latent variables, resulting in a complex, nonlinear manifold in $\mathbb{R}^d$.

### C.3  HEMISPHERE($d'$, $d$) DATASET

The *Hemisphere*($d'$, $d$) dataset consists of samples drawn from the upper hemisphere of a $d'$-dimensional unit sphere, which are then embedded into a $d$-dimensional ambient space using a random isometric embedding. Below are the steps involved in constructing this dataset.

---

**Algorithm 1** General Dataset Generation Algorithm

---

**Require:** Number of samples $N$, MCMC steps $T$, Step size $\delta$, Diffeomorphism $\varphi$, Covariance matrix $\Lambda$

**Ensure:** Dataset $\{\mathbf{x}_1, \mathbf{x}_2, \ldots, \mathbf{x}_N\}$

1: Initialize: Set initial state $\mathbf{x}_0 = \mathbf{0} \in \mathbb{R}^2$.
2: **for** $i = 1$ to $N$ **do**
3:     $\mathbf{x} = \mathbf{x}_0$
4:     **for** $k = 1$ to $T$ **do**
5:         Compute the score function $\nabla_{\mathbf{x}} \log p_{\text{target}}(\mathbf{x})$.
6:         Propose $\mathbf{x}' = \mathbf{x} + \frac{\delta^2}{2}\nabla_{\mathbf{x}} \log p_{\text{target}}(\mathbf{x}) + \delta\boldsymbol{\eta}$, where $\boldsymbol{\eta} \sim \mathcal{N}(\mathbf{0}, \mathbf{I}_2)$.
7:         Compute the forward kernel:

$$K_{\text{forward}} = \frac{|\mathbf{x} - \mathbf{x}' + \frac{\delta^2}{2}\nabla_{\mathbf{x}'} \log p_{\text{target}}(\mathbf{x}')|^2}{2\delta^2}$$

8:         Compute the reverse kernel:

$$K_{\text{reverse}} = \frac{|\mathbf{x}' - \mathbf{x} + \frac{\delta^2}{2}\nabla_{\mathbf{x}} \log p_{\text{target}}(\mathbf{x})|^2}{2\delta^2}$$

9:         Compute the Metropolis-Hastings acceptance probability:

$$A = \min\left(1, \frac{p_{\text{target}}(\mathbf{x}')}{p_{\text{target}}(\mathbf{x})} \exp\left(-K_{\text{forward}} + K_{\text{reverse}}\right)\right)$$

10:        Accept $\mathbf{x}'$ with probability $A$; else set $\mathbf{x}' = \mathbf{x}$.
11:        Update $\mathbf{x} = \mathbf{x}'$.
12:     **end for**
13:     Store the final $\mathbf{x}$ as sample $\mathbf{x}_i$.
14: **end for**

---

**1. Sampling from the Upper Hemisphere**   We begin by sampling points from the upper hemisphere of the $d'$-dimensional unit sphere $S_+^{d'} \subset \mathbb{R}^{d'+1}$. The upper hemisphere is defined as:

$$S_+^{d'} = \left\{ \mathbf{x} \in \mathbb{R}^{d'+1} : \|\mathbf{x}\| = 1, \, x_1 \geq 0 \right\}.$$

The first angular coordinate $\theta_1$ is sampled from a Beta distribution with shape parameters $\alpha = 5$ and $\beta = 5$, scaled to the interval $\left[0, \frac{\pi}{2}\right]$. This sampling method emphasizes points near the "equator" of the hemisphere. The remaining angular coordinates $\theta_2, \ldots, \theta_{d'}$ are sampled uniformly from the interval $[0, \pi]$:

$$\theta_1 \sim \text{Beta}(5, 5) \cdot \left(\frac{\pi}{2}\right), \quad \theta_i \sim \text{Uniform}(0, \pi), \text{ for } i = 2, \ldots, d'.$$

**2. Conversion to Cartesian Coordinates**   Next, each sampled point in spherical coordinates is converted into Cartesian coordinates in $\mathbb{R}^{d'+1}$ using the following transformation equations:

$$x_1 = \cos(\theta_1), \quad x_2 = \sin(\theta_1)\cos(\theta_2), \quad \ldots, \quad x_{d'+1} = \sin(\theta_1)\sin(\theta_2)\cdots\sin(\theta_{d'}).$$

This conversion ensures that the sampled points lie on the surface of the unit sphere in $(d' + 1)$-dimensional space.

**3. Random Isometric Embedding into $\mathbb{R}^d$**   After sampling points on the hemisphere in $\mathbb{R}^{d'+1}$, the points are embedded into a $d$-dimensional ambient space ($d \geq d' + 1$) using a random isometric embedding. The embedding process is as follows:

1. Generate a random matrix $\mathbf{A} \in \mathbb{R}^{d \times (d'+1)}$, where each entry is sampled from a standard normal distribution $\mathcal{N}(0, 1)$.

2. Perform a QR decomposition on matrix $\mathbf{A}$ to obtain $\mathbf{Q} \in \mathbb{R}^{d \times (d'+1)}$:

$$\mathbf{A} = \mathbf{QR}.$$

The columns of $\mathbf{Q}$ form an orthonormal basis for a $(d'+1)$-dimensional subspace of $\mathbb{R}^d$, ensuring that $\mathbf{Q}$ defines an isometric embedding from $\mathbb{R}^{d'+1}$ into $\mathbb{R}^d$. This guarantees that distances and angles are preserved during the mapping, maintaining the geometric structure of the original space within the higher-dimensional ambient space.

3. Use matrix $\mathbf{Q}$ to map each sample $\mathbf{x} \in \mathbb{R}^{d'+1}$ into the ambient space:

$$\mathbf{y} = \mathbf{Q}\mathbf{x},$$

where $\mathbf{y} \in \mathbb{R}^d$ are the embedded samples.

---

**Algorithm 2** Hemisphere$(d', d)$ Dataset Generation

---

1: **Input:** Intrinsic dimension $d'$, ambient dimension $d$, number of samples $n$, Beta distribution parameters $\alpha = 5$, $\beta = 5$
2: **Output:** Dataset $\mathbf{Y} \in \mathbb{R}^{n \times d}$
3: **Step 1: Generate Random Isometric Embedding**
4: Generate a random matrix $\mathbf{A} \in \mathbb{R}^{d \times (d'+1)}$ with entries from $\mathcal{N}(0, 1)$
5: Perform QR decomposition on $\mathbf{A}$ to obtain $\mathbf{Q} \in \mathbb{R}^{d \times (d'+1)}$:

$$\mathbf{A} = \mathbf{Q}\mathbf{R}$$

6: **Step 2: Construct Dataset**
7: **for** $i = 1$ to $n$ **do**
8:     **Step 2.1: Sample Spherical Coordinates**
9:     Sample the first angular coordinate $\theta_1$ from a scaled Beta distribution:

$$\theta_1 \sim \text{Beta}(\alpha, \beta) \cdot \left(\frac{\pi}{2}\right)$$

10:     Sample the remaining angular coordinates $\theta_2, \ldots, \theta_{d'}$ from a uniform distribution:

$$\theta_i \sim \text{Uniform}(0, \pi), \quad \text{for } i = 2, \ldots, d'$$

11:     **Step 2.2: Convert to Cartesian Coordinates**
12:     Convert the spherical coordinates to Cartesian coordinates $\mathbf{x}_i \in \mathbb{R}^{d'+1}$ using:

$$x_1 = \cos(\theta_1), \quad x_2 = \sin(\theta_1)\cos(\theta_2), \ldots, \quad x_{d'+1} = \sin(\theta_1)\sin(\theta_2)\cdots\sin(\theta_{d'}).$$

13:     **Step 2.3: Embed Sample $\mathbf{x}_i$ into Ambient Space**
14:     Map the sample $\mathbf{x}_i$ to the ambient space using:

$$\mathbf{y}_i = \mathbf{Q}\mathbf{x}_i$$

15:     Append $\mathbf{y}_i$ to the dataset $\mathbf{Y}$
16: **end for**
17: **Return:** The final dataset $\mathbf{Y} = [\mathbf{y}_1, \mathbf{y}_2, \ldots, \mathbf{y}_n]$

---

## C.4 SINUSOID$(d', d)$ DATASET

The *Sinusoid*$(d', d)$ dataset represents a $d'$-dimensional manifold embedded in $d$-dimensional space through nonlinear sinusoidal transformations. Below are the detailed steps involved in constructing this dataset.

**1. Sampling Latent Variables** The latent variables $\mathbf{z} \in \mathbb{R}^{d'}$ are sampled from a multivariate Gaussian distribution with zero mean and isotropic variance, as follows:

$$\mathbf{z} \sim \mathcal{N}\left(0, \sigma_m^2 I_{d'}\right),$$

where $\sigma_m^2$ controls the variance along each intrinsic dimension, and $I_{d'}$ is the $d' \times d'$ identity matrix. The value of $\sigma_m^2$ is set to 3 for our experiments.

**2. Defining Ambient Coordinates with Sinusoidal Transformations** For each of the $d - d'$ ambient dimensions, we construct a shear vector $\mathbf{a}_j \in \mathbb{R}^{d'}$, with its elements drawn uniformly from the interval $[1, 2]$:

$$\mathbf{a}_j \sim \mathrm{Uniform}(1, 2)^{d'}, \quad \text{for } j = 1, \ldots, d - d'.$$

The shear vectors $\mathbf{a}_j$ apply a fixed linear transformation to the latent space $\mathbf{z} \in \mathbb{R}^{d'}$, determining how the latent variables influence each ambient dimension. These vectors, sampled once for each of the $d - d'$ ambient dimensions, modulate the scale and periodicity of the sinusoidal transformation.

Each ambient coordinate $x_j$ is generated as a sinusoidal function of the inner product between $\mathbf{a}_j$ and $\mathbf{z}$, with a small Gaussian noise added for regularization.

$$x_j = \sin\left(\mathbf{a}_j^\top \mathbf{z}\right) + \epsilon_j,$$

where $\epsilon_j \sim \mathcal{N}(0, \sigma_a^2)$ is Gaussian noise with variance $\sigma_a^2$. In our experiments, we set $\sigma_a^2 = 10^{-3}$.

**3. Constructing the Dataset Samples** The final samples $\mathbf{y} \in \mathbb{R}^d$ are formed by concatenating the ambient coordinates $x_1, x_2, \ldots, x_{d-d'}$ with the latent variables $z_1, z_2, \ldots, z_{d'}$:

$$\mathbf{y} = [x_1, x_2, \ldots, x_{d-d'}, z_1, z_2, \ldots, z_{d'}]^\top.$$

---

**Algorithm 3** Sinusoid($d'$, $d$) Dataset Generation

---

1: **Input:** Intrinsic dimension $d'$, ambient dimension $d$, number of samples $n$, variance $\sigma_m^2 = 3$, noise variance $\sigma_a^2 = 10^{-3}$
2: **Output:** Dataset $\mathbf{Y} \in \mathbb{R}^{n \times d}$
3: **Step 1: Generate Shear Vectors**
4: **for** $j = 1$ to $d - d'$ **do**
5:     Sample shear vector $\mathbf{a}_j \in \mathbb{R}^{d'}$ from $\mathrm{Uniform}(1, 2)^{d'}$
6: **end for**
7: **Step 2: Construct Dataset**
8: **for** $i = 1$ to $n$ **do**
9:     **Step 2.1: Sample Latent Variables**
10:     Generate latent variables $\mathbf{z}_i \in \mathbb{R}^{d'}$ from a multivariate Gaussian:

$$\mathbf{z}_i \sim \mathcal{N}(0, \sigma_m^2 \cdot I_{d'})$$

11:     **Step 2.2: Compute Ambient Coordinates for Sample** $i$
12:     **for** $j = 1$ to $d - d'$ **do**
13:         Compute ambient coordinate $x_j$ for the $i$-th sample:

$$x_j = \sin\left(\mathbf{a}_j^\top \mathbf{z}_i\right) + \epsilon_j, \quad \epsilon_j \sim \mathcal{N}(0, \sigma_a^2)$$

14:     **end for**
15:     **Step 2.3: Form Final Sample** $\mathbf{y}_i$
16:     Concatenate the ambient coordinates $\mathbf{x} = [x_1, x_2, \ldots, x_{d-d'}]$ and the latent variables $\mathbf{z}_i$ to form the final sample $\mathbf{y}_i \in \mathbb{R}^d$:

$$\mathbf{y}_i = [x_1, x_2, \ldots, x_{d-d'}, z_1, z_2, \ldots, z_{d'}]^\top$$

17:     Append $\mathbf{y}_i$ to the dataset $\mathbf{Y}$
18: **end for**
19: **Return:** The final dataset $\mathbf{Y} = [\mathbf{y}_1, \mathbf{y}_2, \ldots, \mathbf{y}_n]$

---

## D    ERROR METRICS FOR EVALUATION OF PULLBACK GEOMETRIES

**Geodesic Error.** The geodesic error measures the difference between geodesics on the learned and ground truth pullback manifolds. Given two points $\mathbf{x}_0, \mathbf{x}_1 \in \mathbb{R}^d$, let $\gamma_{\mathbf{x}_0, \mathbf{x}_1}^{\varphi_{\theta_2}}(t)$ and $\gamma_{\mathbf{x}_0, \mathbf{x}_1}^{\varphi_{\mathrm{GT}}}(t)$ denote

the geodesics induced by the learned map $\varphi_{\theta_2}$ and the ground truth map $\varphi_{\text{GT}}$, respectively, where $t \in [0, 1]$.

The geodesic error is calculated as the mean Euclidean distance between the learned and ground truth geodesics over $N$ pairs of points:

$$\text{Geodesic Error} = \frac{1}{N} \sum_{i=1}^{N} \frac{1}{T} \sum_{k=1}^{T} \left\| \gamma^{\varphi_{\theta_2}}_{\mathbf{x}_0^{(i)}, \mathbf{x}_1^{(i)}}(t_k) - \gamma^{\varphi_{\text{GT}}}_{\mathbf{x}_0^{(i)}, \mathbf{x}_1^{(i)}}(t_k) \right\|_2,$$

where $T$ is the number of time steps used to discretize the geodesic, and $t_k = \frac{k-1}{T-1}$ for $k = 1, \ldots, T$.

This metric captures the average discrepancy between the learned and ground truth geodesics, reflecting the accuracy of the learned pullback manifold.

**Variation Error.** The variation error quantifies the sensitivity of the geodesic computation under small perturbations to one of the endpoints. For two points $\mathbf{x}_0, \mathbf{x}_1 \in \mathbb{R}^d$, let $\mathbf{z} = \mathbf{x}_1 + \Delta\mathbf{x}$, where $\Delta\mathbf{x}$ is a random variable sampled from the Gaussian distribution:

$$\Delta\mathbf{x} \sim \mathcal{N}(\mathbf{0}, 0.1^2\mathbf{I}),$$

with mean $\mathbf{0}$ and covariance $0.1^2\mathbf{I}$, where $\mathbf{I}$ is the identity matrix. Define $\gamma^{\varphi_{\theta_2}}_{\mathbf{x}_0, \mathbf{x}_1}(t)$ and $\gamma^{\varphi_{\theta_2}}_{\mathbf{x}_0, \mathbf{z}}(t)$ as the geodesics from $\mathbf{x}_0$ to $\mathbf{x}_1$ and $\mathbf{z}$, respectively, induced by the learned map $\varphi_{\theta_2}$.

The variation error is calculated as the mean Euclidean distance between the geodesic from $\mathbf{x}_0$ to $\mathbf{x}_1$ and the perturbed geodesic from $\mathbf{x}_0$ to $\mathbf{z}$:

$$\text{Variation Error} = \frac{1}{N} \sum_{i=1}^{N} \frac{1}{T} \sum_{k=1}^{T} \left\| \gamma^{\varphi_{\theta_2}}_{\mathbf{x}_0^{(i)}, \mathbf{x}_1^{(i)}}(t_k) - \gamma^{\varphi_{\theta_2}}_{\mathbf{x}_0^{(i)}, \mathbf{z}^{(i)}}(t_k) \right\|_2,$$

where $N$ is the number of sampled point pairs, $T$ is the number of time steps used to discretize the geodesic, and $t_k = \frac{k-1}{T-1}$ for $k = 1, \ldots, T$.

This metric evaluates the robustness of the learned geodesic against small perturbations, providing insight into the stability of the learned manifold.

# E  TRAINING DETAILS

The following section describes the important configuration parameters for reproducing the experiments on manifold mappings. All experiments share some common parameters, which are listed below, while dataset-specific parameters are provided in Table 2.

**Common Parameters:**

- **Optimizer:** Adam with `betas = (0.9, 0.99)`, `eps = 1 × 10^{-8}`, and weight decay of $1 \times 10^{-5}$.

- **Learning Rate Schedule:** Warm-up cosine annealing with 1000 warm-up steps.

- **Gradient Clipping:** Gradient norm clipped to 1.0.

- **Model Architecture:** A composition of affine coupling layers is used, where each layer transforms part of the input while keeping the other part unchanged. The transformation function in each layer is modeled by a residual network (ResNet) consisting of 64 hidden features, 2 residual blocks, ReLU activations, and no batch normalization. Dropout is set to 0, and transformations alternate across different dimensions at each layer.

Table 2: Training configurations for each experiment.

| Dataset | Flow Steps | Epochs | Batch Size | $\lambda_{\text{iso}}$ | $\lambda_{\text{vol}}$ | Learning Rate |
|---|---|---|---|---|---|---|
| Sinusoid(1,3) | 8 | 1000 | 64 | 1.0 | 1.0 | $3 \times 10^{-4}$ |
| Sinusoid(2,3) | 8 | 1000 | 64 | 1.0 | 1.0 | $3 \times 10^{-4}$ |
| Sinusoid(5,20) | 24 | 2000 | 128 | 1.2 | 2.5 | $4 \times 10^{-4}$ |
| Hemisphere(2,3) | 8 | 2000 | 64 | 1.0 | 1.0 | $4 \times 10^{-4}$ |
| Hemisphere(5,20) | 12 | 2000 | 64 | 0.75 | 1.2 | $4 \times 10^{-4}$ |

## F  DATA MANIFOLD APPROXIMATION

The learned manifold, shown in orange in Figure 1, is the set $D_\epsilon(\mathcal{U})$, where $D_\epsilon$ is the RAE decoder (21), the set $\mathcal{U}$ in the latent space is the open set given by

$$\mathcal{U} = \prod_{i=1}^{d_\epsilon} (-3\sqrt{\mathbf{a}_{u_i}}, 3\sqrt{\mathbf{a}_{u_i}})$$

and $\mathbf{a}_{u_1}, \ldots, \mathbf{a}_{u_{d_\epsilon}}$ are the $d_\epsilon$ highest learned variances corresponding to the ones used in the RAE construction.

To visualize this in practice, we construct a mesh grid by linearly sampling each latent dimension from $-3\sqrt{\mathbf{a}_{u_i}}$ to $+3\sqrt{\mathbf{a}_{u_i}}$, for $i = 1, \ldots, d_\epsilon$, where $d_\epsilon$ is the number of significant latent dimensions. Practically, the off-manifold latent dimensions (those corresponding to negligible variances) are set to zero. The decoder $D_\epsilon$ then maps this grid from $\mathcal{U}$ back to $\mathbb{R}^d$, generating an approximation of the data manifold, as illustrated in Figure 1.

## G  EXPERIMENTS WITH MORE COMPLEX DISTRIBUTIONS

We applied our training framework[9] to model complex real-world and synthetic distributions, specifically focusing on the subset of digit "1" from the MNIST dataset and a synthetic dataset of 10-dimensional Gaussian blobs introduced in Stanczuk et al. (2022). The subset of digit "1" is chosen as it is likely to be represented well by the unimodal parametric family 3. The Gaussian blobs dataset is included because its intrinsic dimension is known (10), providing a reliable baseline for evaluating the accuracy of the RAE's intrinsic dimension estimation.

Modeling such distributions effectively requires more expressive normalizing flow architectures, such as affine coupling flows combined with $1 \times 1$ invertible convolutions for pixel reshuffling, or rational quadratic (RQ) spline flows. These architectures, however, are not guaranteed to have zero second derivatives, which can cause the higher-order terms in Theorem 1 to become significant, potentially inflating the expected reconstruction error of the Riemannian Auto-encoder (RAE). Furthermore, enforcing $\ell^2$ isometry regularization becomes more challenging in these cases.

Despite these issues, our experiments indicate that the deviations from isometry and the presence of non-zero second derivatives do not visibly impact the quality of the manifold mappings. However, they can affect the overall performance of the RAE.

We trained two models on the digit "1" subset of MNIST: an affine coupling flow with $1\times1$ invertible convolution layers (which is not an affine transformation) and an RQ spline flow. In both cases, we observed stable and accurate geodesics that traversed regions of high data density, consistent with theoretical predictions. These geodesics effectively navigate through common examples of the digit "1", as expected based on the learned data distribution. The results are presented in Figure 7.

To complement the MNIST experiments, we evaluated the same models on the Gaussian blobs dataset, where the true intrinsic dimension is known to be 10. This dataset allows us to directly assess the accuracy of the RAE's intrinsic dimension estimation. The trained affine and RQ spline models produced stable and accurate geodesics similar to those observed in the MNIST experiments, as shown in Figure 8. However, both models overestimated the intrinsic dimension.

---

[9]with the minor change of replacing the isometry regularizer to a more scalable version (see Appendix H for details)

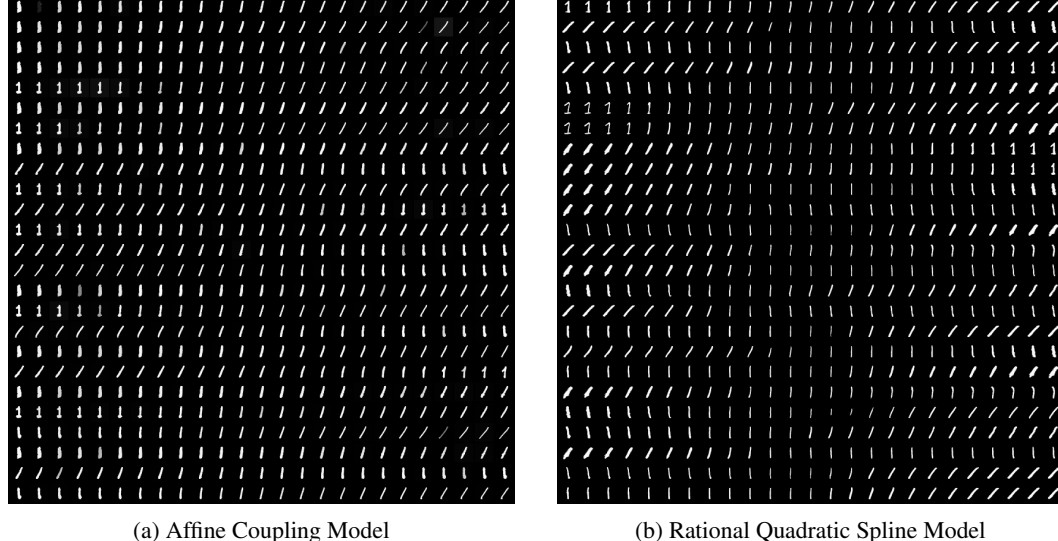

(a) Affine Coupling Model        (b) Rational Quadratic Spline Model

Figure 7: Geodesics computed for two different normalizing flow models trained on the subset of digit "1" from the MNIST dataset. (a) shows the geodesics for a model using affine coupling layers with $1 \times 1$ convolutions, while (b) shows the geodesics for a model using rational quadratic splines. In both cases, the geodesics pass through high-density regions of the dataset, consistent with the theoretical expectation that geodesics align with areas of higher probability under the learned data distribution. This highlights the models' ability to capture the underlying data manifold effectively.

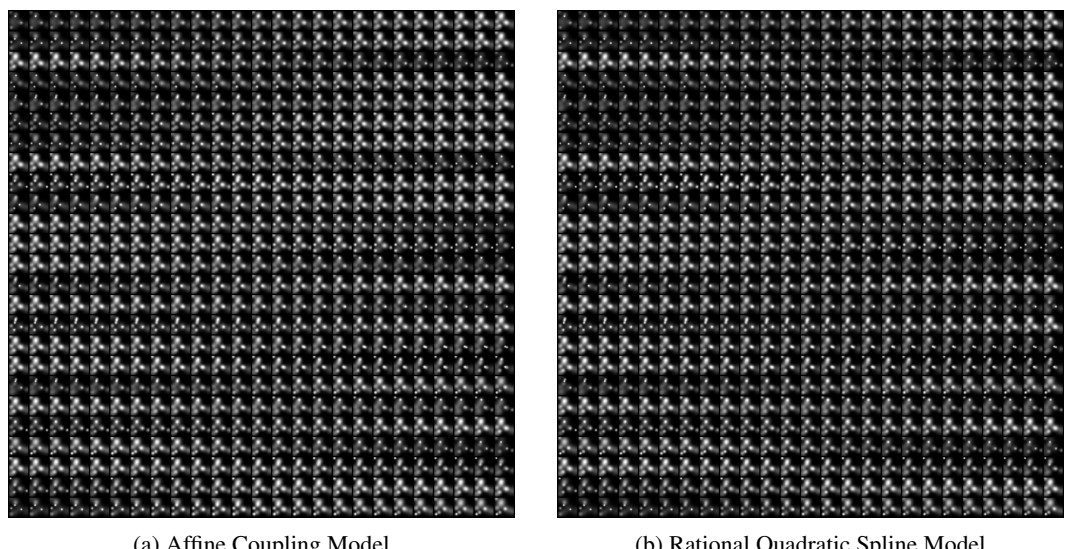

(a) Affine Coupling Model        (b) Rational Quadratic Spline Model

Figure 8: Geodesics computed for two different normalizing flow models trained on the 10-dimensional Gaussian blobs dataset. (a) shows the geodesics for a model using affine coupling layers with $1 \times 1$ convolutions, while (b) shows the geodesics for a model using rational quadratic splines. Both models demonstrate smooth geodesics that traverse high-density regions.

Our RAE model consistently overestimates the intrinsic dimension across both datasets. For the MNIST subset, we observe an estimated intrinsic dimension of approximately 650 for the RQ spline flow and around 300 for the affine flow when using an $\epsilon = 0.1$ threshold. Similarly, for the Gaussian blobs dataset, the affine model estimates an intrinsic dimension of 650, while the RQ spline model estimates 396. We attribute this overestimation primarily to the difficulty of achieving an $\ell^2$ isometry while learning the complex data distribution. Although non-zero second derivatives are

a secondary factor, they may exacerbate the issue by increasing the contributions of higher-order terms in Theorem 1.

These results suggest that while our method can effectively capture the manifold structure, additional regularization may be required to better align the learned metric with the true geometry, especially when using highly expressive flow architectures.

## H  COMPUTATIONAL COMPLEXITY OF THE PROPOSED APPROACH TO TRAINING

In this paper we have claimed that this approach is more scalable than the work by Diepeveen (2024). This is the case for most parts of the proposed loss, except for the isometry regularizer, which is also in the loss by Diepeveen (2024).

In our work, we employed the **exact orthogonal regularization**, which comes down to computing

$$
\frac{1}{b} \sum_{i=1}^{b} \|(D_{\mathbf{x}^i}\varphi_{\theta_2})^\top D_{\mathbf{x}^i}\varphi_{\theta_2} - \mathbf{I}_d\|_F^2,
$$

where $b$ is the batch size.

**Computational Complexity**  The complexity of the exact method is:

$$
O(b \times d^3 + b \times d \times f),
$$

where:

- $d$ is the ambient dimension,
- $f$ is the cost of a forward and backward pass through $\varphi$.

This scales **cubically with** $d$ and is **independent of the intrinsic dimension**. We leveraged PyTorch's `vmap` to efficiently compute this for dimensions up to $d = 100$ in our experiments.

**Approximate Method for Higher Dimensions**  In the experiments for higher-dimensional data (in appendix G), we used an **approximate regularization method**. Instead of computing the full Jacobian, we approximate the orthogonality condition using $v$ random orthonormal vectors $\{\mathbf{v}^j\}_{j=1}^v$. The regularization term is

$$
\frac{1}{b} \sum_{i=1}^{b} \sum_{j=1}^{v} \left\|(D_{\mathbf{x}^i}\varphi_{\theta_2})^\top D_{\mathbf{x}^i}\varphi_{\theta_2}[\mathbf{v}^j] - \mathbf{v}^j\right\|^2.
$$

**Complexity of Approximate Method**

$$
O(b \times d \times v^2 + b \times v \times f + b \times v^3).
$$

This reduces the computational cost, scaling **linearly with** $d$, and is also independent of the intrinsic dimension. It offers a scalable alternative for high-dimensional datasets. For our main experiments, we used the exact method due to its strong regularization in moderate dimensions ($d \leq 100$). However, the approximate method was tested in preliminary high-dimensional experiments and effectively enforced orthogonality, promoting near-isometric mappings as required by our theoretical framework. The exact method ensures robust regularization in lower to moderate dimensions, while the approximate method provides a scalable alternative for higher-dimensional cases. By leveraging a small number of slicing vectors, it reduces the computational burden while preserving key geometric properties, making it effective across varying dimensional regimes.

