# OpenReview forum: "Score-based pullback Riemannian geometry"
_ICLR.cc/2025/Conference — Submitted to ICLR 2025_

### Official Review · Reviewer_y5Bx · 2024-11-03

**Soundness:** 2
**Presentation:** 1
**Contribution:** 2
**Rating:** 5
**Confidence:** 3

**Summary:**

The authors propose a metric between data samples based on the score information of probability density. Many notations are used without clear definitions. I can assume p in line 135 represents a point in the data manifold, and the score is a vector in the tangent space according to Eq. (5). The tangent space similarity determines the metric between sample points. Because the probability density in use adopts the form of energy-based model with a convex energy psi, the equations can be formulated using the gradient of psi. Along with the learning objective function from normalizing flow (NF), a data generating algorithm can be developed. When an NF is employed, the NF neural network is the diffeomorphism mentioned in the text. It is unclear how the metric obtained from D_X\phi_{\theta_2} information makes sense.

**Strengths:**

The notion of point similarity using Riemanian metric from generative models is interesting. The algorithm works well with synthetic data upon complex manifold.

**Weaknesses:**

The authors need to provide clear explanations of how the equations are motivated and how the algorithms can be implemented.
Real-world experiment is missing. The authors made a strong manifold assumption, and it is necessary to know how the algorithm works with data having noisy manifold.
What do the contour lines in Figure 2 represent? There is no information what the horizontal and vertical axes represent nor the meaning of the contour lines in the figure.
The paper lacks motivation for why the score information should be used in the metric. Why not simply use \nabla p?

**Questions:**

Can you provide the motivation why tangent space similarity obtained from density function is related to metric?
Can you provide a detailed procedure for learning?
The figures should be explained better.

---

> ### Author Response · Authors · 2024-11-18
>
> • Strengths:
>
> – We thank the reviewer for highlighting the interesting connection
> between Riemannian geometry and generative modeling!
>
> • Weaknesses
>
> – Regarding the first item:
>
> ∗ Would the reviewer be more specific which equations should be
> motivated more clearly?
>
> ∗ For section 3, we would like to note that the main motivation for
> all equations come from the fact that these lead to closed-form
> manifold mappings in proposition 1. We also state in section 3
> that this result is the main motivation for considering the metric
> tensor field of interest in the first place and give intuition (and
> now a more general proof in proposition 2 in the appendix) for
> why geodesics and other manifold mappings will move through
> the data support.
>
> ∗ For section 4, all equations are motivated by the definition of a
> Riemannian autoencoder given by Diepeveen (2024). The specific
> way of constructing it is motivated by Theorem 1, which is
> also motivated from a high level in this section.
>
> ∗ For section 5, the loss follows directly once we decide that we
> want to train using normalizing flows, but want to take the theory
> into account (which gives the regularizers).
>
> – Regarding the second item:
>
> ∗ We agree with the reviewer that this was currently not in the
> paper. We provide results for MNIST in the revised version
> (appendix G). We need a more expressive architecture for this,
> which makes it harder to regularize for isometry. In the experiment
> the geodesics are visually good, but the dimension is still
> off, since we do not satisfy isometry.
>
> ∗ As we also wrote to other reviewers, this is an obstacle that can
> be overcome and is ongoing work that is beyond the scope of this
> paper.
>
> – Regarding the third item:
>
> ∗ We thank the reviewer for bringing this up. The method is actually
> by design incorporating noise. The whole idea is that we find
> the whole distribution (which is noisy) and interpolate through
> it or select only the dimensions in the RAE that have largest
> variance (so the non-noise dimensions).
>
> ∗ So our method does not directly fit a manifold and move over
> it, but we can interpolate between any points in space (even if
> starting at some noisy data point) and for the RAE select an
> error margin ϵ which controls the dimension of the manifold we
> will use to approximate the data distribution. We would like
> to highlight that we start the paper with this, i.e., “Data often
> reside near low-dimensional non-linear manifolds as illustrated
> in Figure 1.” In this figure (and for any of the other experiments
> or toy examples) the data is noisy.
>
> – Regarding the fourth item:
>
> ∗ These are the level sets of a distribution of the form we are
> interested in. This is just a toy example. The axes don’t carry
> any meaning, which is why there are no labels
>
> ∗ We added this in the figure caption
>
> – Regarding the fifth item:
>
> ∗ We agree that from first glance it is hard to see why this metric
> tensor field should be considered. As also stated in the reply for
> the first item, the motivation for considering this metric tensor
> field is just that we get geodesics of the right form that move
> through the data distribution. There is no further reason to
> attach too much meaning to the metric tensor field apart from
> the geometry it generates.
>
> ∗ If we would use ∇p, we would get a singular metric tensor field
> at the suprema of the distribution. So this would not be a good
> choice to start with. On top of that, even if regularization were
> to be used, there are no closed-form mappings for geodesics etc.,
> which is the main motivation for using the proposed Riemannian
> structure (as stated in section 3).
>
> ∗ We do understand this might be unsatisfying for the reviewer.
> So we added an extra result (proposition 2) to the appendix,
> which should make the picture a bit more complete. It states
> (informally) that this type of geometry gives under conditions
> on psi geodesics (and therefore other manifold mappings) that
> move through the data distribution.
>
> • Questions
>
> – Regarding the first question:
>
> ∗ Assuming that the reviewer means $(Ξ, Φ)^{∇ψ◦φ}_x$ , this is the metric
> tensor field that generates the Riemannian structure. The
> Riemannian geometry on one’s space of interest (in this work
> just Rd) is generated from just this object. Once the metric
> tensor field is chosen, all manifold mappings (including the metric/
> distance) are defined. This is all written out in section 2.
>
> – Regarding the second question:
>
> ∗ For learning, you minimize the loss in section 5. All details
> for every experiment (hyper-parameters etc.) are provided in
> appendices. The source code is already available, but the link is
> not in the paper for anonymity reasons. Could the reviewer be
> more specific what is still missing?
>
> – Regarding the third question:
>
> ∗ We fixed Figure 2. Are there any other figures that need more
> explanation? Overall we feel that we are fairly verbose in the
> figure captions. If the reviewer disagrees, it would be great if
> they could specify which figures would need improvement.

---

> > ### Author Response · Authors · 2024-11-25
> >
> > We hope that our reply clarifies and alleviates the reviewer’s concerns. If this is the case, we kindly ask the reviewer to consider raising their rating, given that they are acknowledging the novelty, the strengths and the contributions of our paper.

---

### Official Review · Reviewer_5u2b · 2024-11-04

**Soundness:** 3
**Presentation:** 3
**Contribution:** 2
**Rating:** 5
**Confidence:** 2

**Summary:**

This work introduces score-based pullback Riemannian geometry, a scalable framework that combines elements of pullback Riemannian geometry and generative models to enhance interpretable representation learning. Focusing on unimodal distributions, it features a Riemannian autoencoder (RAE) with closed-form geodesics that traverse the data's probability density, allowing for accurate estimation of the data manifold's dimension. Numerical experiments show that the framework effectively produces high-quality geodesics and reliably assesses intrinsic dimensions, even in high-dimensional spaces.

**Strengths:**

1.	This framework facilitates the learning of interpretable representations by leveraging Riemannian geometry, which can capture complex data structures more effectively than traditional methods.
2.	Under the assumption of the unimodal distribution, the score-based approach facilitates the construction of the pullback geometry with closed-form manifold mappings.
3.	The paper also showed that the resulting geodesics always pass through the support of data probability

**Weaknesses:**

1.	The proposed approach has a strong assumption on the data distribution (unimodal distributions), which limits its applicability to more complex or multimodal data.
2.	In the experimental results, the proposed method is only compared with some discrete-time normalizing flow methods (NF, Anisotropic NF, Isometric NF). It is too limited.

**Questions:**

1. What is the computational complexity of the proposed approach? How does it scale with the intrinsic dimension of the data dimension and the ambient dimension?
2. What is the main difficulty to extend it to multimodal data?

---

> ### Author Response · Authors · 2024-11-18
> **[1/2]**
>
> • Strengths:
>
> – We thank the reviewer for highlighting the interpretability our method
> offers!
>
> • Weaknesses:
>
> – Regarding the first item (see the related question by reviewer WnXv
> and QfsL):
>
> ∗ We agree that the way in which the work is presented, the setup
> seems limited. Having that said, in training we can actually already
> have multiple modes because we train it as an adapted
> NF (and in practice we have visibly good geodesics for more
> complicated data sets such as MNIST digit 1). For the downstream
> geometry we could just use the learned diffeomorphism
> and strongly convex function. One of the main points of the
> paper is that from theory we would expect that without modifications
> we might be unable to guarantee stability of manifold
> mapping and find the right dimension with the RAE. We can
> confirm that we indeed get too high a dimension as we would
> hope for in the case of MNIST although we get visibly good
> geodesics. We added this experiment to the appendix.
>
> ∗ Without disclosing too much on ongoing research, there are actually
> two ways we are working on to work around current practical
> challenges: a more complicated base distribution (e.g., multimodal
> Gaussian along the lines of [1]) or post-processing under
> a less restricted normalizing flow (so that det is not equal to 1).
> Either way, additional theory is needed to get this to work and
> there are several extra factors that need to be regularized for,
> both of which are beyond the scope of this work. So we decided
> not to include it, but we are happy to disclose that we are actively
> working on this (and to mention that this can be made
> computationally feasible!).
>
> ∗ Having that said, we would like to emphasize that the subsequent
> projects really build on top of the insights from this paper. So
> we feel strongly that having this base case well-understood will
> give a good intuition for downstream work to build upon it as
> we can very clearly state what it can and cannot do.
>
> ∗ Finally, to showcase to the reader that multimodality is in principle
> included in the way the NF training is set up, we added a
> footnote in section 5.
>
> – Regarding the second item:
>
> ∗ We agree with the reviewer that it would have been nice to compare
> to more methods. Having that said, the point of the first
> experiment is to showcase that the adaptations to normalizing
> flow are needed to get decent geometry.
>
> ∗ For the remainder of the experiments (on RAE performance) it
> would have made sense to compare to other methods. However,
> there is to the best of our knowledge no work on Riemannian
> geometries in which all manifold mappings is (or even can be)
> computed in an efficient and stable way. In particular, if there
> is no closed-form solution, you would need very high-accuracy
> numerical solvers to find approximate manifold mapping to such
> that the exp and the log are approximate inverses. In our experience,
> the only consistent way of doing this is by [2], which is not
> used in the cited ML works. In these works only geodesics are
> considered and the way these are computed, there is typically
> little regard for solving the geodesic equation consistently and
> in a stable fashion (let along accurately). Hence, constructing
> RAEs in the sense of [3] would be very unstable (and very slow)
> for any other method, which is why we have not compared to
> any other mehod for this second part.
>
> [1] Izmailov, Pavel and Kirichenko, Polina and Finzi, Marc and Wilson,
> Andrew Gordon, Semi-supervised learning with normalizing flows, International
> conference on machine learning, 4615–4630, 2020
>
> [2] Rumpf, Martin and Wirth, Benedikt, Variational time discretization of
> geodesic calculus, IMA Journal of Numerical Analysis, 2015
>
> [3] Diepeveen, Willem, Pulling back symmetric Riemannian geometry for
> data analysis, arXiv preprint arXiv:2403.06612, 2024

---

> ### Author Response · Authors · 2024-11-18
> **[2/2]**
>
> • Questions
>
> – Regarding the first question: Please allow us to break this question
> down into two parts
>
> ∗ Computational Complexity of the Proposed Approach
> In our work, we employed the exact orthogonal regularization,
> defined as:
> $$\frac{1}{b} \sum_{i=1}^b\\|J_i^{\top} J_i-I\\|_F^2$$
> where:
>
> · Ji is the Jacobian of the transformation ϕ for the i-th sample,
>
> · I is the identity matrix,
>
> · b is the batch size.
>
> Computational Complexity The complexity of the exact
> method is:
>
> $$O(b × d^3 + b × d × f),$$
>
> where:
>
> · d is the ambient dimension,
>
> · f is the cost of a forward and backward pass through ϕ.
>
> This scales cubically with d and is independent of the intrinsic
> dimension. We leveraged PyTorch’s vmap to efficiently
> compute this for dimensions up to d = 100 in our experiments.
>
> ∗ Approximate Method for Higher Dimensions In preliminary
> experiments for higher-dimensional data, we used an approximate
> regularization method. Instead of computing the
> full Jacobian, we approximate the orthogonality condition using
> v random orthonormal vectors $(v_j, v_j)_2$
> =1. The regularization term
> is:
>
> $$ \frac{1}{b} \sum_i^b \sum_{j=1}^v\\|J_i^{\top} J_i v_j-v_j\\|^2 \text {, }$$
>
> where $J_i v_j$ is the Jacobian-vector product for sample i and vector
> $v_j$ .
>
> Complexity of Approximate Method
>
> $$O(b × d × v^2 + b × v × f + b × v^3).$$
>
> This reduces the computational cost, scaling linearly with d,
> and is also independent of the intrinsic dimension. It offers a
> scalable alternative for high-dimensional datasets. For our main
> experiments, we used the exact method due to its strong regularization
> in moderate dimensions (d ≤ 100). However, the approximate
> method was tested in preliminary high-dimensional
> experiments and effectively enforced orthogonality, promoting
> near-isometric mappings as required by our theoretical framework.
> The exact method ensures robust regularization in lower
> to moderate dimensions, while the approximate method provides
> a scalable alternative for higher-dimensional cases. By leveraging
> a small number of slicing vectors, it reduces the computational
> burden while preserving key geometric properties, making it effective
> across varying dimensional regimes.
>
> – Regarding the second question:
>
> ∗ Please see the response of first item in weaknesses. In a nutshell:
> theory and additional regularization, both of which go beyond
> the scope of this work.

---

> > ### Author Response · Authors · 2024-11-25
> >
> > We hope that our reply clarifies and alleviates the reviewer’s concerns. If this is the case, we kindly ask the reviewer to consider raising their rating, given that they are acknowledging the novelty, the strengths and the contributions of our paper.

---

> > ### Comment · Reviewer_5u2b · 2024-11-27
> >
> > Thank you to the authors for their detailed responses and additional results. I have a follow-up question:
> >
> > 1. For the computational complexity, do you mean $O(b\times d^3 + b \times d \times f)$ for the exact method and $O(b \times d \times v^2 + b \times v \times f + b \times v^3)$? The current expressions in the response is not very clear. Could the authors include the discussion on the computational complexity in the revised paper?
> >
> > I think my concern regarding the practical application of the proposed method remains. Appendix G shows the results on MNIST with a single digit, which has a much simpler distribution compared with the whole dataset. Additionally, the overestimation of dimensionality for the Gaussian blobs in Appendix G raises some concerns. As a result, I will maintain my current score for now.

---

> > > ### Author Response · Authors · 2024-11-27
> > >
> > > We completely agree with the reviewer that the focus of this paper is on learning transformations towards unimodal distributions. However, we want to emphasize that this is a foundational step towards addressing the more complex challenge of multimodal distributions. What is more, this *does not* mean that multimodal distributions cannot be learned. The overestimation for Gaussian blobs most likely occurs precisely due to this reason, i.e., the true distribution is in fact not unimodal, leading to an overestimation in dimension, however it still results in accurate geodesics. So when the reviewer writes "Additionally, the overestimation of dimensionality for the Gaussian blobs in Appendix G raises some concerns", can the reviewer be more specific on their exact concern? It is in line with theory that in case that we cannot get the dimension right if need more expressivity from the model to capture the geometry (which we can as shown by the correct geodesics).
> > >
> > > We want to clearly emphasise that *scalable* and *accurate* calculation of geodesics and manifold mappings, especially in high dimensions is currently not something that exists (as we highlighted in the introduction). This paper directly addresses this gap by introducing a novel approach that achieves both scalability and accuracy for unimodal distributions (we added the discussion on the complexity to appendix H and corrected the typo the reviewer pointed out). We believe this contribution is substantial, particularly for the computational geometry community where existing methods often struggle with the scale of problems we tackle.
> > >
> > > We have outlined clear next steps for extending our method to multimodal distributions and for correcting for lack of isometry (so that we get a more accurate estimation of the dimension), which we are actively pursuing. We believe a phased approach, starting with a robust foundation for the unimodal case, is crucial for achieving a reliable and effective solution for the more general multimodal scenario.
> > >
> > > Finally, the reviewer writes that "my concern regarding the practical application of the proposed method remains". Please allow us to argue for the applicability of the method. As a practical application of this line of work, we would like to point the reviewer to [1] (a follow-up on work by Diepeveen (2024), which heavily inspired this work). In [1], the authors showcase that having good geodesics (from pullback geometry) is already a very strong geometric prior for generation and outperforms state-of-the-art methods like conditional flow matching (notice that in our case we would use the adapted normalizing flow training mainly to learn the geometry and use a more sophisticated type of diffusion model to actually get higher-quality samples). The authors of [1] also show that any dimension reduction (so it does not have to be tight) already helps on top of that. So being able to get rid of 2/3 of the dimensions for MNIST or the blobs is expected to boost downstream performance drastically. The current limitation of [1] is that is has been build on top of Diepveen (2024), which is limited to small data sets as already discussed in our paper. We hope that this gives the reviewer a better idea the potential future impact of this paper. We added this reference [1] to the introduction to convey this message better.
> > >
> > >
> > > [1] de Kruiff, Friso and Bekkers, Erik and ¨Oktem, Ozan and Sch¨onlieb, Carola-Bibiane and Diepeveen, Willem, Pullback Flow Matching on Data Man- ifolds, arXiv preprint arXiv:2410.04543, 2024

---

### Official Review · Reviewer_QfsL · 2024-11-07

**Soundness:** 3
**Presentation:** 3
**Contribution:** 3
**Rating:** 6
**Confidence:** 3

**Summary:**

In the article, the authors introduce the scored-base Riemannian structure that utilizes pullback Riemannian geometry in conjunction with generative models. Specifically:
(1) Based on Gaussian-like distribution in the Euclidean space, they define a Riemannian metric, which is linked to the pullback of the score function.
(2) A comprehensive study about this metric is provided, including the geodesics, logarithmic map, exponential map, distance, and barycenter.
(3) They propose a novel auto-encoder and decoder methods, based on the metric, offering theoretical guarantees on the consistency of estimation (Theorem 1).
(4) A mechanism for learning probability densities is introduced, utilizing an adapted normalizing flow loss function, with multiple experiments.

**Strengths:**

In overall, the approach in this article is both innovative and compelling. Specifically:
1. The Riemannian metric structure proposed in the article is notably original, with a comprehensive examination of its fundamental geometric properties.
2. The proof supporting auto-encoder and decoder mechanism is well-articulated and appears to be mathematically rigorous.
3. The paper is clearly written.

**Weaknesses:**

1. The data used in the study is somewhat artificial. The work would benefit from employing more realistic datasets.
2. Although the methodology is quite straightforward, it would be preferable if the authors included step-by-step derivations to verify the  geometric properties (mentioned in Proposition 1).
3. Given that the data is generated from quadratic structures, the model appears to perform well only for data with similar characteristics (such as the banana-shaped data in the paper). For more complex and diverse data structures, the model may require further development to capture such complexities.

**Questions:**

1. As I mentioned in the weakness section, although the quasi-Gaussian assumption of the data is comprehensible, can we generalize it to another class of density functions?
2. Do we have any computational advantage when using this method comparing with the other ones (for ex, NF, Anisotropic NF, Isometric NF)?
3.  Can you elaborate on the intuition of using the score function in this article?

---

> ### Author Response · Authors · 2024-11-18
>
> • Strengths
>
> – We thank the reviewer for their kind words, in particular for emphasizing
> the originality of this work!
>
> • Weaknesses
>
> – Regarding the first item: We conducted additional experiments on
> more realistic datasets, including the digit ’1’ subset of MNIST and
> the Gaussian Blobs dataset. While our method produced high-quality
> geodesics, the Riemannian Autoencoder (RAE) tended to overestimate
> the intrinsic manifold dimension. These results, along with a
> detailed discussion, are now included in the newly added Section G
> of the Appendix in the revised manuscript.
>
> – Regarding the second item: We agree that the reader could use some
> help with the details to verify proposition 1. We have added the
> proof proof to the appendix. It should be easier for the reader to go
> through the steps while having the right references directly there.
>
> – Regarding the third item (see the related question by reviewer WnXv
> and 5u2b):
>
> ∗ Indeed, the current setup focuses on quadratic structures. However,
> there are actually two ways to make it more realistic: a
> more complicated base distribution (e.g., multimodal Gaussian
> along the lines of [1]) or post-processing under a less restricted
> normalizing flow (so that det is not equal to 1), which also gives
> multi-modality. Either way, additional theory is needed to get
> this to work and there are several extra factors that need to be
> regularized for, both of which are beyond the scope of this work.
> So we decided not to include it, but we are happy to disclose
> that we are actively working on this (and to mention that this
> can be made computationally feasible!).
>
> ∗ Having that said, we would like to emphasize that the subsequent
> projects really build on top of the insights from this paper. So
> we feel strongly that having this base case well-understood will
> give a good intuition for downstream work to build upon it as
> we can very clearly state what it can and cannot do.
>
> • Questions
>
> – Regarding the first question:
>
> ∗ We agree that the way in which the work is presented, the setup
> seems limited. Having that said, in training we can actually already
> have multiple modes because we train it as an adapted NF
> (and in practice we have visibly good geodesics for more complicated
> data sets such as MNIST as mentioned above). For the
> downstream geometry we could just use the learned diffeomorphism
> and strongly convex function. One of the main points
> of the paper is that from theory we would expect that without
> modifications we might be unable to guarantee stability of manifold
> mapping and find the right dimension with the RAE. We
> can confirm that we indeed get too high a dimension as we would
> hope for in the case of MNIST digit 1 although we get visibly
> good geodesics. We added this experiment to the appendix.
>
> ∗ As already mentioned above, there are several ways of generalizing
> the theory to accommodate for multimodality without
> losing out on comutational feasibility, but these are beyond the
> scope of this work. So rather soon, we believe that the seemingly
> shortcomings of this paper can be alleviated.
>
> ∗ Finally, to showcase to the reader that multimodality is in principle
> included in the way the NF training is set up, we added a
> footnote in section 5.
>
> – Regarding the second question:
>
> ∗ We would first like to highlight that our method aims to learn
> Riemannian geometry while standard NF methods aim to generate
> data! Having that said, we don’t have computational advantage
> when using them our method compared to the other ones.
> It’s more computationally expensive than (NF and Anisotropic
> NF), as we need to compute the isometry regularisation term.
> However, through our experiments we proved that it’s necessary
> to combine isometry regularisation with base distribution
> anisotropy in order to obtain more accurate and stable manifolds
> maps as evidenced by the significantly improved geodesic
> and variation error. The adapted framework additionally allows
> for the construction of an interpretable latent space, which is not
> possible with the other variations.
>
> – Regarding the third question (see the related question by reviewer
> WnXv):
>
> ∗ We would like to highlight that from just looking at the metric
> tensor field it will be hard to see why using information related to
> the score is a good choice. However, as we already explain in section
> 3, the motivation of the chosen structure comes solely from
> proposition 1. We also provide interpretations of what geodesics
> behave like in the paragraph that start with “This special case
> highlights why”.
>
> [1] Izmailov, Pavel and Kirichenko, Polina and Finzi, Marc and Wilson,
> Andrew Gordon, Semi-supervised learning with normalizing flows, International
> conference on machine learning, 4615–4630, 2020

---

> > ### Author Response · Authors · 2024-11-25
> >
> > We hope that our reply clarifies and alleviates the reviewer’s concerns. If this is the case, we kindly ask the reviewer to consider raising their rating, given that they are deeply acknowledging the novelty, the strengths and the contributions of our paper.

---

### Official Review · Reviewer_WnXv · 2024-11-09

**Soundness:** 2
**Presentation:** 2
**Contribution:** 2
**Rating:** 5
**Confidence:** 3

**Summary:**

This paper introduces a score-based pullback Riemannian geometry—a data-driven geometry in which geodesics follow the data supports. Assuming specific unimodal densities, it derives closed-form expressions for geodesics, distances, the exponential map, and the logarithmic map, and formulates a Riemannian autoencoder with an error bound. Additionally, it proposes a generative model by adapting normalizing flow as a learning algorithm. The model is tested on synthetic datasets, demonstrating its capability to compute geodesics along the data manifold, estimate intrinsic dimensions, and provide global coordinate charts.

**Strengths:**

- The proposed geometry seems novel, with intriguing closed-form solutions derived from the geometry.
- Most of the derivations seem correct (I could not verify all the details).

**Weaknesses:**

- The properties of the proposed geometry are insufficiently discussed and lack intuitive explanations. For example, it would be helpful to explain how the geodesics behave and the form of the Riemannian metrics for different distributions. Without such clarification, the current title may overstate the paper’s contributions.
- Practical applications of the proposed geometry, beyond the Riemannian autoencoder, are not clearly identified. For the Riemannian autoencoder, it remains unclear in what contexts it outperforms alternative methods.
- Furthermore, the unimodal density assumption is restrictive; because computational efficiency depends heavily on this assumption, extending the method to multimodal cases while retaining computational advantages seems challenging.
- The experiments are limited to synthetic data, without comparisons to similar Riemannian geometry methods, such as those mentioned in the introduction (e.g., Arvanitidis et al. 2016).

**Questions:**

In Table 1, how are the starting and goal points for geodesics, as well as the perturbations, selected when evaluating the geodesic and variation errors?

---

> ### Author Response · Authors · 2024-11-18
> **[1/2]**
>
> • Strenghts
>
> – We thank the reviewer for highlighting the novelty of our work!
>
> • Weaknesses
>
> – Regarding the first item (see the related question by reviewer QfsL):
>
> ∗ We agree that just from looking at the metric tensor field it will
> be hard to see why this is a good choice. However, as we already
> explain in section 3, the motivation of the chosen structure comes
> solely from proposition 1. We also provide interpretations of
> what geodesics behave like in the paragraph that start with “This
> special case highlights why”. We invite the reviewer to specify
> what other properties of the geometry they want discussed.
>
> ∗ One can also derive directly what the resulting metric tensor is -
> which is precisely the squared hessian of the log likelihood. Hessian
> metrics have been widely studied in the literature, however
> here by simply squaring the matrix we are able to recover exact
> manifold mappings - fundamentally not altering the metric
> itself. In particular, this is always positive (semi)-definite and
> will always attract geodesics towards stationary points - which
> in the case of unimodal distributions is unique.
>
> ∗ We also agree that the intuition for the proposed geometry in the
> original version (discussed in the paragraph starting with “This
> special case highlights why”) mainly focuses on the Gaussian
> case. A more general case can be stated and proven, and is
> provided in the appendix (proposition 2). This new result gives
> conditions for when geodesics (and manifold mappings for that
> matter) will pass through the data support.
>
> – Regarding the second item:
>
> ∗ We agree that this can be emphasized. In the initial version
> of the paper we have tried to do this by showcasing how widely
> used VAEs are and what the potential of Riemannian geometry is
> given all the methods out there that have been generalized to the
> manifold setting. However, in the paper we did not have a lot of
> space to elaborate on the shortcomings of VAEs in the mentioned
> scientific applications nor space to use pullback geometry for a
> concrete downstream application. Please allow us to discuss this
> here.
>
> · Regarding shortcomings of VAEs: Very often people try to
> attach meaning to the VAE latent space (comparing distances,
> discussing densities or computing interpolants), which
> is a very problematic practice. For example, in Zhong (2021)
> cited in the paper, latent space interpolation is used to visualize
> protein trajectories. While this is nice to get a feeling
> of what has been learned, it is hard to make any scientific
> claims on how proteins change from one conformation to the
> next. Similar arguments can be made for the other papers
> cited.
>
> · Regarding pullback geometry in downstream applications:
> Also when using VAEs for sampling, key properties might
> not be preserved for this reason, but which can be salvaged
> in a pullback setting. As a recent example consider [1], a
> follow up work on the work by Diepeveen (2024) (which has
> been the inspiration for this paper as well), in which this
> case is actually showcased for the downstream application of
> generation.
>
> ∗ So to emphasize the potential future impact of our methods we
> added the sentence “After that, we believe that this line of work
> has wide variety of downstream applications as many of the applications
> mentioned to motivate this line of work will benefit
> from more interpretable representation learning.” at the end
> of the conclusion. This way there is a nice connection back to
> the introduction, which should give the reader a reminder of the
> broader impact of this paper.
>
> [1] de Kruiff, Friso and Bekkers, Erik and ¨Oktem, Ozan and Sch¨onlieb,
> Carola-Bibiane and Diepeveen, Willem, Pullback Flow Matching on Data Man-
> ifolds, arXiv preprint arXiv:2410.04543, 2024

---

> ### Author Response · Authors · 2024-11-18
> **[2/2]**
>
> – Regarding the third item (see the related question by reviewer QfsL
> and 5u2b):
>
> ∗ We agree that the way in which the work is presented, the setup
> seems limited. Having that said, in training we can actually already
> have multiple modes because we train it as an adapted NF
> (and in practice we have visibly good geodesics for more complicated
> data sets such as MNIST digit 1, which is not expected to
> be a perfect deformed Gaussian). For the downstream geometry
> we could just use the learned diffeomorphism and strongly convex
> function. One of the main points of the paper is that from
> theory we would expect that without modifications we might
> be unable to guarantee stability of manifold mapping and find
> the right dimension with the RAE. We indeed can confirm that
> the recovered dimension is too high in the case of MNIST, despite
> visibly good geodesics. We added this experiment to the
> appendix.
>
> ∗ Without disclosing too much on ongoing research, there are actually
> two ways we are working on to work around current practical
> challenges: a more complicated base distribution (e.g., multimodal
> Gaussian along the lines of [2]) or post-processing under
> a less restricted normalizing flow (so that det is not equal to 1).
> Either way, additional theory is needed to get this to work and
> there are several extra factors that need to be regularized for,
> both of which are beyond the scope of this work. So we decided
> not to include it, but we are happy to disclose that we are actively
> working on this (and to mention that this can be made
> computationally feasible!).
>
> ∗ Having that said, we would like to emphasize that the subsequent
> projects really build on top of the insights from this paper. So
> we feel strongly that having such a “restrictive case” will give a
> good intuition for downstream work to build upon it as we can
> very clearly state what it can and cannot do.
>
> ∗ Finally, to showcase to the reader that multimodality is in principle
> included in the way the NF training is set up, we added a
> footnote in section 5.
>
> – Regarding the fourth item:
>
> ∗ The method actually also works on single digits of MNIST (we
> didnt have time to check for the whole data set), but we only
> get good geodesics, not the right dimension. This can also be
> fixed in post-processing, but that would be beyond the scope
> of this work (as mentioned above). A detailed discussion and
> presentation of the geodesics has been added in Appendix G in
> the revised version of the paper.
>
> • Question:
>
> – The starting and goal points for geodesics are randomly sampled
> from the test data. For the variation error, a perturbed point z is
> created by adding a small random Gaussian perturbation to the goal
> point x1, defined as z = x1+0.1·N(0, I). This detail is now explicitly
> included in Appendix D of the revised manuscript.
>
> [2] Izmailov, Pavel and Kirichenko, Polina and Finzi, Marc and Wilson,
> Andrew Gordon, Semi-supervised learning with normalizing flows, International
> conference on machine learning, 4615–4630, 2020

---

> > ### Author Response · Authors · 2024-11-25
> >
> > We hope that our reply clarifies and alleviates the reviewer’s concerns. If this is the case, we kindly ask the reviewer to consider raising their rating, given that they are acknowledging the novelty, the strengths and the contributions of our paper.

---

> > > ### Comment · Reviewer_WnXv · 2024-12-01
> > >
> > > I appreciate the authors’ clarifications and additional experiments.
> > >
> > > However, it remains intuitively unclear why the Euclidean metric on the score vector space should be pulled back via the score vector mapping. Specifically, why is the squared Hessian of the log-likelihood considered the metric tensor? Does it even satisfy the coordinate transformation rule of a metric tensor? Providing intuitive explanations or examples, such as visualizations of Riemannian metrics in the data space, could help clarify this concept.
> > >
> > > Regarding the comparison to VAE, it would be beneficial if the advantages were explicitly demonstrated through experiments in the paper.
> > >
> > > While I understand that the multimodal setting is expected to be addressed in future extensions, the current work still has limitations in this regard.
> > >
> > > I will slightly increase my score based on these considerations.

---

> > > > ### Author Response · Authors · 2024-12-01
> > > >
> > > > >  However, it remains intuitively unclear why the Euclidean metric on the score vector space should be pulled back via the score vector mapping. Specifically, why is the squared Hessian of the log-likelihood considered the metric tensor? Does it even satisfy the coordinate transformation rule of a metric tensor?
> > > >
> > > > The overall line of thought for understanding why this is the metric is as follows:
> > > > 1. There is no *unique* perfect metric given a density.
> > > > 2. We wish to put a metric which is both *computationally not expensive* and *geodesics go through high likelihood regions*.
> > > > 3. We highlight that for most methods, e.g. Sorrenson et al., have a metric which admits good geodesics, but is *expensive*.  Most of the time *expensive* is the issue.
> > > > 4. Our theoretical contribution (Proposition 2) shows that indeed *geodesics go through high likelihood regions*, while Proposition 1 shows that they are indeed *computationally not expensive*.
> > > >
> > > > In terms of coordinate transformations - it is a fairly standard result that a pullback tensor field generates a Riemannian structure through the Jacobian of the transformation, thus indeed it is a tensor. See e.g. John M. Lee - Introduction to Smooth Manifolds [Proposition 13.9]. Basically, to show that it is a tensor, you need to show that it is a $C^\infty(M)$ module (so linear in both coordinates wrt smooth functions). This is true as multiplication of a tangent vector with a smooth function can be taken outside of the bracket for this inner product.
> > > >
> > > > > Providing intuitive explanations or examples, such as visualizations of Riemannian metrics in the data space, could help clarify this concept.
> > > >
> > > > We are happy to provide an example for the final version, as it is no longer possible to update the paper, specifically deriving this metric in the case of the exponential family and visualising them. Proposition 1 and 2 directly apply to these and we agree that they could be very useful for understanding what the resulting metric tensor is.
> > > >
> > > > > Regarding the comparison to VAE, it would be beneficial if the advantages were explicitly demonstrated through experiments in the paper.
> > > >
> > > > Thank you for the suggestion. While we agree that a direct comparison would be informative, previous studies e.g. [1] have extensively explored the limitations of VAEs in this regard, and more generally similar issues with latent space interpolation were seen in prior work [2].
> > > >
> > > > [1] de Kruiff, Friso and Bekkers, Erik and ¨Oktem, Ozan and Sch¨onlieb, Carola-Bibiane and Diepeveen, Willem, Pullback Flow Matching on Data Manifolds, arXiv preprint arXiv:2410.04543, 2024
> > > > [2] Georgios Arvanitidis, Lars K Hansen, and Søren Hauberg. A locally adaptive normal distribution. Advances in Neural
> > > > Information Processing Systems, 29, 2016.
> > > >
> > > > > I will slightly increase my score based on these considerations.
> > > >
> > > > We thank the reviewer once again for their useful feedback and highlighting the novelty of our work, and appreciate your commitment to raising the score. We would appreciate if you may be able to do this before the end of the discussion period!

---

### Author Response · Authors · 2024-11-27

Dear Area Chair,

We extend our sincere gratitude for your invaluable oversight of the review process. The dedication shown by you and the reviewers in evaluating our manuscript is deeply appreciated. The feedback received has been instrumental in enhancing our research presentation and refining our core contributions.

As the discussion period continues, we respectfully request your assistance in encouraging further reviewer engagement. Currently, we've received feedback from only one reviewer, and there has been limited discussion regarding our improvements. We are eager to address any concerns and participate in a meaningful dialogue to ensure comprehensive consideration of all feedback.

Your guidance in facilitating this interaction would be immensely valuable. We believe that increased engagement could significantly enhance our submission and contribute to a fair and thorough evaluation of our work.

Best regards,

The authors

---

### Meta-Review · Area_Chair_GJyH · 2024-12-21

**Metareview:**

In the paper, the authors proposed score-based pullback Riemannian geometry, a generative data-driven framework that integrates Riemannian geometry.

While the reviewers agree that the proposed method is quite interesting and intuitive, there are several weaknesses of the current paper: (1) the computational efficiency depends heavily on the unimodal density assumption, which is restrictive. Adapting the current method to handle multimodal cases while maintaining computational advantages seems challenging. (2) The experimental results are limited to only synthetic data. It is unclear about the practical performance of the proposed method in the real-world settings. (3) The properties of the proposed geometry are insufficiently discussed and not intuitive. Furthermore, the writing of the current paper is also not clear.

Given the above concerns raised by the reviewers which have not been fully addressed after the rebuttal, I recommend rejecting the paper at the current stage. I encourage the authors to include the suggestions and feedback of the reviewers into the revision of their manuscript.

**Additional Comments On Reviewer Discussion:**

Please refer to the metareview.

---

### Decision · Program_Chairs · 2025-01-22

Reject